

# Can remote sensing combustion phase improve estimates of landscape fire smoke emission rate and composition?

Farrer Owsley-Brown[1,2,3], Martin J. Wooster[1,2,3], Mark J. Grosvenor[1,2,3], Yanan Liu[1,2]

[1]Department of Geography, King's College London, Bush House, 30 Aldwych, London, WC2B 4BG
[2]Leverhulme Centre for Wildfires, Environment and Society, King's College London, London, WC2R 2LS
[3]NERC National Centre for Earth Observation, King's College London, London, WC2R 2LS

*Correspondence to*: Farrer Owsley-Brown (farrer.owsley-brown@kcl.ac.uk)

**Abstract.** The proportion of flaming and smoldering activity occurring in landscape fires varies with fuel type and fuel characteristics, which themselves are influenced by ecology, meteorology, time since the last fire etc. The proportion of

these combustion phases greatly influences the rate of fuel consumption and smoke emission, along with the chemical composition of the smoke, which influences the effects on the atmosphere. Earth Observation (EO) has long been suggested as a way to remotely map combustion phase, and here we provide the first known attempt at evaluating whether such approaches can lead to the desired improvements in smoke emissions estimation. We use intensively measured laboratory burns to evaluate two EO approaches hypothesized to enable remote determination of combustion phase and concurrent

measurements of the smoke to determine how well each is able to improve estimation of smoke emission rates, smoke composition and the overall rate of fuel consumption. The first approach aims to estimate the sub-pixel 'effective fire temperature', which has been suggested to differ between flaming and smoldering combustion, and the second detects the potassium emission line (K-line) believed only to be present during flaming combustion. We find while the fire effective temperature approach can be suited to estimating Fire Radiative Power (FRP), it does not significantly improve on current

approaches to estimate smoke chemical makeup and smoke emission. The K-line approach does however provide these improvements when combined with the FRP data, improving the accuracy of the estimated $CO_2$ emission rate by an average of $17\pm4\%$ and $42\pm15\%$, respectively, depending on whether the K-line detection is used to simply classify the presence of flaming combustion, or whether its magnitude is also used to estimate its relative proportion. Estimates of CO and $CH_4$ emission rates were improved to a lesser extent than that of $CO_2$, but the accuracy of the smoke modified combustion

efficiency (MCE) estimates increased by $30\pm15\%$ and $46\pm10\%$, respectively. MCE is correlated to the emissions factors

(EFs) of many smoke constituents, so remotely deriving MCE provides a way to tailor these during smoke emissions calculations. Whilst we derived and tested our approaches on laboratory burns, we demonstrate their wider efficacy using airborne EO data of a boreal forest wildfire where we find that combined used of K-line and FRP data significantly change estimated smoke MCE and $CO_2$ and CO emission rates compared to the standard approach. Our findings suggest that

satellite EO methods that jointly provide K-Line and FRP data could enable marked improvements in the mapping of landscape fire combustion phase, fuel consumption and smoke emissions rate and composition.

## 1. Introduction

Satellite Earth Observation (EO) is the only approach able to provide global, systematic, and regularly repeated estimates of landscape fire trace gas and aerosol emissions (Chuvieco et al., 2019; Wooster et al., 2021). 'Bottom-up' EO-based

emissions estimation approaches rely on calculating the amount of biomass burned, based on burned area (BA) or fire radiative energy (FRE) measures (Giglio et al., 2013; Kaiser et al., 2012; Wiedinmyer et al., 2011), which is then multiplied by the emission factor ($EF_x$) of the chemical species (X) of interest. $EF_x$ represents the mass of that species that is emitted per unit of fuel burned (g kg$^{-1}$; Andreae and Merlet, 2001). More recently developed EO-based approaches now link EO-derived fire radiative power (FRP) measures directly to emission rates of a trace gas or particulate species via an emission coefficient

$C_e^X$ (kg MJ$^{-1}$; Ichoku and Kaufman, 2005; Mota and Wooster, 2018; Nguyen et al., 2023; Nguyen and Wooster, 2020). However, no EO approach currently takes into account combustion phase and the proportion of flaming and smoldering activity, despite this being known to dramatically influence fire and smoke characteristics (e.g., Freeborn et al., 2008; Urbanski, 2014; Zhang et al., 2015).

The proportion of flaming and smoldering combustion in a landscape fire depends on fuel type and fuel condition – for

example being influenced by fuel load, fuel density and fuel moisture content for example (e.g., Burling et al., 2010; Garg et al., 2024; Urbanski, 2013). It is well known that for most chemical species released by landscape burning, the $EF_x$ (and the $C_e^x$) change markedly between the flaming and smoldering combustion phase (e.g., Reid et al., 2005; Zhang et al., 2015). Therefore, since the proportion of flaming and smoldering combustion varies even between fires burning in the same fuel



type, smoke emissions characteristics can also vary widely – which then has an impact on the fire's effects on atmospheric

composition (e.g., Mebust and Cohen, 2013; Zheng et al., 2018).

Biome-specific databases of $EF_X$ and/or $C_e^x$ do not generally report separate values for flaming and smoldering combustion, but rather overall 'fire-averaged' values based on laboratory and/or field measurements assumed to include a 'typical' amount of flaming and smoldering combustion for that fuel type (see Akagi et al., 2011; Andreae, 2019; Andreae and Merlet, 2001). Remote sensing measures of FRP reflect an instantaneous observation, possibly at a time when the amount of

flaming and smoldering combustion may be very atypical of the 'average'. Furthermore, even 'fire-averaged' emission factors and coefficients likely vary between fires in the same fuel. For these reasons there is increasing interest in providing more dynamic emission factors and emissions coefficients, initially at least to cope with their presumed seasonal variations (Vernooij et al., 2023, 2022).

There have long been suggestions that remote sensing may be able to provide ways to specify more tailored emissions

factors or coefficients (e.g. Andreae and Merlet, 2001; Freeborn et al., 2008; Kaufman et al., 1998). The most common approach suggested uses estimates of sub-pixel fire effective temperature, for example derived via the Dozier (1981) approach or similar multispectral analysis methods that can be used with ground or airborne data (e.g. Dennison et al., 2006) but also with spaceborne data to analyze fires covering only a very small fraction of the pixel area (e.g., Giglio and Kendall, 2001). An alternative approach would be to remotely identify a phenomena characteristic of only a single combustion phase,

and most commonly suggested is the potassium line (K-line) radiative emission associated only with flaming activity (Amici et al., 2011; Magidimisha and Griffith, 2017; Vodacek et al., 2002). The current work aims to test these remote sensing approaches to determine whether they really can improve smoke emissions rate and smoke composition estimation, in this case, of the three most dominant trace gases ($CO_2$, CO and $CH_4$). We use a series of intensively instrumented combustion chamber burns for this, and also demonstrate the best approaches on real landscape fires using airborne EO data – providing

a demonstration that is of intermediate scale to satellite EO.



## 2. Background

Though the burning of biomass involves multiple different combustion phases, almost all the smoke is produced in the flaming and smoldering phases (Bertschi et al., 2003; Yokelson et al., 1997). The flaming phase generally involves higher temperatures, and the resulting oxidation of fuel carbon is typically far more complete and leads to a higher $CO_2$ emission

factor Zhang et al., 2015). EFs of species such as $NO_X$, $SO_2$ and black carbon (BC) are also elevated in the flaming phase (Andreae and Merlet, 2001; Reid et al., 2005). Smoldering combustion commonly refers to a combination of pyrolysis (thermal decomposition of the fuel) and glowing combustion (of char), mostly involving lower temperatures and more incomplete oxidation of the fuel carbon compared to flaming. $CO_2$ EFs are lower, but those of species such as CO, $CH_4$, organic carbon (OC), and volatile organic compounds (VOCs) are higher (e.g. Zhang et al., 2015; Andreae, 2019; Bertschi et

al., 2003; Reid et al., 2005; Yokelson et al., 1997, 1996). However, fuel combustion rate is also important to consider in emissions rate calculations, and is far higher per unit area for flaming rather than smoldering combustion (e.g., Lacaux et al., 1996; Wooster et al., 2011) . Furthermore, despite $EF_{CO2}$ being typically 5-15% lower for smoldering than flaming combustion, smoldering fires still release the majority of their carbon as $CO_2$ as the emissions factor of $CO_2$ is still higher than for any other compound (e.g., Reisen et al., 2018; Zhang et al., 2015). However, the EF of 'preferentially smoldering'

compounds such as CO, $CH_4$, organic carbon (OC), and VOCs are typically many times higher during smoldering than flaming phase combustion, so along with the fuel consumption rate the amount of smoldering activity plays a very significant influence in their emission rate.

Whilst landscape fires often show periods of only smoldering combustion, periods of 'flaming-dominated' activity are, typically, relatively short. Far more common are stages with a mixed contribution, where some smoldering is happening

behind the flaming front (e.g., Bertschi et al., 2003; Burling et al., 2011; Rabelo et al., 2004; Urbanski, 2014; Yokelson et al., 1997). Since combustion rate per unit area is generally far higher in the flaming-dominated phase than the smoldering-dominated phase, the total production of even 'preferentially smoldering' species can often be higher during the flaming period rather than during pure smoldering, depending on the area affected by each and their relative durations. This complexity has led to metrics like the modified combustion efficiency (MCE), which aims to quantify the balance between

flaming and smoldering combustion in the smoke production process:



$$MCE = \frac{\Delta CO_2}{\Delta CO_2 + \Delta CO} \tag{1}$$

where $\Delta X$ indicates the excess concentration of $CO_2$ or CO (commonly measured in ppmv).

Purely flaming combustion results in an MCE close to 1.0 due to minimal CO production, whilst purely smoldering combustion can yield smoke with an MCE as low as 0.65 (depending on fuel type; Akagi et al., 2011; Andreae, 2019). Smoke MCE has been shown to be negatively correlated with the EFs of many preferentially smoldering species (Bertschi et al., 2003; McMeeking et al., 2009; Urbanski, 2013; Yokelson et al., 1996), and understanding the MCE of the fire can therefore enable more precision to be placed on the resulting fire emissions. However, collecting MCE data on landscape fires is challenging even using *in situ* aircraft sampling due to atmospheric mixing (Yokelson et al., 2013), and whilst satellite EO has shown an ability to probe smoke emissions ratios (e.g., Coheur et al., 2009; Ross et al., 2013) it has not yet been possible to remotely sense smoke MCE. Currently, therefore, remote sensing approaches potentially able to determine the amounts of flaming and smoldering combustion ongoing in a fire are based either on retrieving the fire's effective temperature (e.g., Zhukov et al., 2006) or detection of the flaming-phase K-line signature (e.g., Amici et al., 2011).

Remotely sensed fire effective temperature estimation was first proposed by Dozier (1981). Observations in two different wavebands are used to retrieve a fire's subpixel effective temperature ($T_r$) and proportional area (p), with the fire assumed to be thermally homogeneous and superimposed on a thermally homogeneous background with temperature ($T_b$) and proportional area (1-p). Blackbody behavior is generally assumed, and estimation of $T_r$ and p is conducted via solution of:

$$L_1 = \tau_1 p B(\lambda_1, T_r) + (1 - p)L_{b,1} + L_{atm,1} \tag{2}$$

$$L_2 = \tau_2 p B(\lambda_2, T_r) + (1 - p)L_{b,2} + L_{atm,2} \tag{3}$$

where $L_1$ and $L_2$ are spectral radiances ($Wm^{-2}sr^{-1}m^{-1}$) in wavebands $\lambda_1$ and $\lambda_2$, $\tau_1$ and $\tau_2$ are the atmospheric transmittances in those wavebands, and $L_{atm,1}$ and $L_{atm,2}$ are the atmospherically emitted radiances measured by the sensor in those wavebands. $L_{b,1}$ and $L_{b,2}$ are the radiance contributions from the non-burning uniform background, whose temperature is generally estimated from neighboring pixels. The spectral emission at wavelength $\lambda$ and Temperature T is given by Planck's Radiation Law:

$$B(\lambda, T) = \frac{2hc^2}{\lambda^5} \frac{1}{e^{\frac{hc}{\lambda k_B T}} - 1} \tag{4}$$



where $h$ is Planck's constant ($6.62607004 \times 10^{-34}$ kgm²s⁻¹), $k_B$ is Boltzmann's constant ($1.38064852 \times 10^{-23}$ kgm²s⁻²K⁻¹) and $c$ the velocity of light in a vacuum ($299792458$ ms⁻¹).

Dennison et al. (2006), Dennison and Matheson (2011); Matheson and Dennison (2012), Zhukov et al. (2006) provide examples of mapping sub-pixel fire effective temperature, which can be expanded to estimate increased thermal component

fits if more than two wavebands are available, such as in hyperspectral data (Dennison et al., 2006; Dennison and Matheson, 2011; Giglio and Justice, 2003; Giglio and Kendall, 2001; Waigl et al., 2019). This could potentially determine separate flaming and smoldering contributions. However, no studies have yet linked such retrievals to an ability to better estimate smoke emission characteristics (Wooster et al., 2021), despite that being a key aim for such data.

The alternative K-line approach is based on the detection of an emission doublet in the near infrared (766.5 nm and 769.9

nm), which is caused by thermally excited potassium atoms within the burning fuel (Vodacek et al., 2002; Amici et al., 2011; Dennison and Roberts, 2009; Magidimisha et al., 2023; Magidimisha and Griffith, 2017). Only flaming phase activity is hot enough to produce K-line emission and Amici et al. (2011) thus far provide its only detection from space – defining the 'Advanced K Band Difference' (AKBD) metric to quantify its strength:

$$\text{AKBD} = \text{Max}|\text{BandK}_i| - \text{Bkg} \qquad\qquad (5)$$

where $\text{Max}|\text{BandK}_i|$ is the maximum spectral radiance recorded in the 764 nm – 772 nm range (the region that encompasses

the NIR K-line doublet) and Bkg is that recorded just outside the K-emission region, for example 779nm. The K-line signature can be seen superimposed on the background Planckian signal in data from our experiment shown in Fig. 2a.

## 3. Laboratory experiment method

### 3.1 Experiment setup

Experiments were conducted at King's College London's Wildfire Testing Chamber, located at Rothamsted Research,

Harpenden (UK). The physical arrangement and instruments are detailed in Fig. 1 and Table 1. The remote sensing instruments were positioned to view the fuel-bed at nadir (location B in Fig. 1) through appropriately transparent windows (also listed in Table 1) with high heat resistance. Pre-experiment calibrations were undertaken to allow the non-unitary transmissivity of the windows to be taken account of during data analysis.



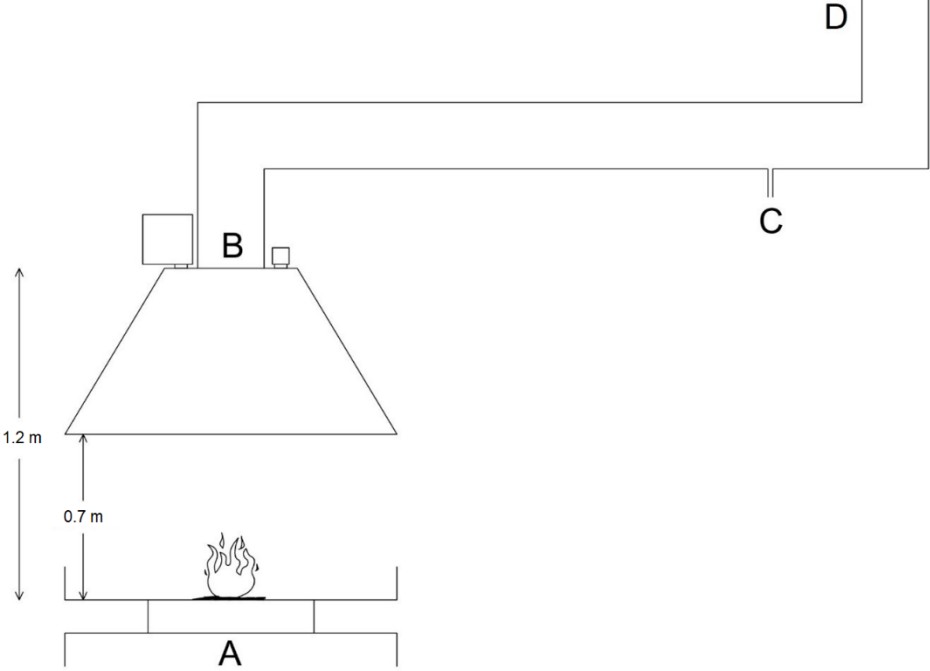

**Figure 1. Experimental setup for examining fire and smoke characteristics: (A) Fuel-bed; (B) Spectrometers and cameras viewing the fuel-bed and nadir through holes in the extraction flue; (C) Gas analyzers, (D) Air flow rate sensor.**

**Table 1. Instruments used and their specifications, as well as the window compositions that protected the spectrometers and cameras viewing through the holes shown in Fig. 1.**

| Location | Instrument | Window | Specifications |
|----------|------------|--------|----------------|
| A | Digital Scales | NA | 0.005 kg readability |



| B | VIS-SWIR spectrometer - SVC HR-1024i | Sapphire (AL$_2$O$_3$) | Spectral Range: 350 nm – 2500 nm <br><br> FWHM: 3.5 nm (700 nm), 9.5 nm (1500 nm), 6.5 nm (2100 nm) <br><br> Bandwidth: 1.5nm (350 nm – 1000 nm), 3.8 nm (1000 nm – 1890 nm), 2.5 nm (1890 nm – 2500 nm) <br><br> FOV: 14° foreoptic lens <br><br> Calibration Accuracy: ±5% (400 nm), ±4% (700 nm), ±7% (2000 nm) <br><br> Measurement frequency: 0.14 – 0.25 Hz |
|---|---|---|---|
| B | Thermal camera - Optris PI 400 | Zinc Selenide (ZnSe) | Spectral Range: 7.5 µm – 13 µm <br><br> Optical Resolution: 382 x 288 pixels <br><br> Framerate: 27 Hz <br><br> Temperature Range: 150 °C – 900 °C <br><br> FOV: 62° x 49° <br><br> f = 8 mm <br><br> Measurement frequency: 1 Hz |
| B | UV-NIR spectrometer - Ocean Insight OCEAN-HDX-XR | Fused silica (SiO$_2$) | Spectral Range: 200 nm – 1100 nm <br><br> FWHM: 1.1 nm <br><br> FOV: 30° fiber-optic guide <br><br> Measurement frequency: 1 Hz |
| B | RGB camera - Apeman A79 | Fused silica (SiO$_2$) | Resolution: 20 MP <br><br> Framerate: 30 fps <br><br> FOV: 170° |





| C | $CO_2$, CO and $CH_4$ analyzer - Los Gatos Research Ultraportable Emissions Analyzer | NA | $CO_2$: 0 – 3000 ppm<br>CO: 0 – 1000 ppm<br>$CH_4$: 0 – 100 ppm<br>$H_2O$: 0 – 99% relative humidity<br>Measurement frequency: 1 Hz |
|---|---|---|---|
| C | $CO_2$ and CO analyzer - GasLab CM-1000 | NA | $CO_2$: 0 – 10000 ppm<br>CO: 0 – 5000 ppm<br>Measurement frequency: 0.5 Hz |
| D | Air pressure differential analyzer - Testo 440 | NA | Pressure differential measuring range: -150 hPa to 150 hPa<br>Accuracy: ±0.2 hPa<br>Resolution: 0.01 hPa<br>Measurement frequency: 1 Hz |

**3.2 Fuel**

Three types of fuel were burned during these experiments: oak kindling, pine forest litter and soybean crop residue. Oak kindling was selected for its relative uniformity, though the thickness and dryness of the individual pieces was not identical and led to some intra-fire variability in the amount of flaming and smoldering activity. The pine forest litter was a mixture of needles, cones and small branches collected from the floor of a UK forest containing mainly Corsican, Maritime, and Scots pine. This fuel mix was dried indoors for a month prior to the burns and the samples burned were selected to maintain the proportions of needles, cones, and branches found on the forest floor. The soybean residues were sourced from China as an example of an agricultural waste product commonly burned in open fields. All fuels were arranged to fit within a 29 cm diameter circle to fit within the measurement area of all remote sensing instruments deployed.



### 3.3 Fire measurements

**3.3.1 Optical and thermal imagery**

A standard RBG camera recorded video imagery of each fire for context, helping gauge how amounts of flaming and smoldering combustion changed over each fire's duration. A calibrated longwave infrared (LWIR) camera (Optris PI400) recorded infrared brightness temperature imagery at 1 Hz, with the 3.3 mm pixel size allowing for an assumption of pixel thermal homogeneity. A 1 Hz FRP record of the fire could then be derived using the Stefan-Boltzmann law with data from each image:

$$FRP = \sum_i \sigma \; a \; T_i^4 \tag{6}$$

where $\sigma$ is the Stefan-Boltzmann constant ($5.670374 \times 10^{-8}$ $Wm^{-2}K^{-1}$), a is the pixel area ($1.109 \times 10^{-5}$ $m^2$), and $T_i$ is the pixel brightness temperature (K). This sum was over the $i$ pixels within each image that had T ³ 600 K, which excluded cooling non-combusting pixels (e.g., Freeborn et al., 2008; Wooster et al., 2011).

**3.3.2 Fire effective temperature and FRP retrieval from spectral fits**

Fire effective temperatures and FRP were also derived using UV-to-SWIR spectral radiance measurements collected at 0.14-0.25 Hz from a calibrated field spectrometer (SVC HR1024i; Table 1). Fire effective temperature estimates, akin to those coming from the Dozier (1981) 'dual-band' approach, were retrieved from the measured spectra ($L_{\lambda, measured}$) – though the use of hyperspectral data enabled three thermal components to be derived (similar to in Wooster et al., 2005; Dennison et al., 2006; and Amici et al., 2011). Modelled spectra ($L_{\lambda, modelled}$) that provided the best fits to the measured spectra were constructed from three individually modelled spectral radiances ($L_{FD}+L_{SD}+L_C$) coming respectively from assumed flaming-dominated (FD, range 923-2000 K), smoldering-dominated (SD, 623-1023 K) and cooling (C, 280-623 K) emitters:

$$L_{\lambda, modelled} = \sum_i^{n=3} p_i \; B_\lambda(T_i) \tag{7}$$

where the fractional areas sum to 1, and $B_\lambda(T_I)$ the Planck radiance emitted by each fractional component was calculated using Eq. 4. Brightness temperature ($T_i$) and fractional area ($p_i$) of each component was iterated to provide the best match between $L_{\lambda, modelled}$ and ($L_{\lambda, measured}$).





A blackbody assumption was made during application of Eq. 7, and burns were conducted in the dark to prevent contamination by reflected sunlight. Due to the small distances involved in the measurement setup, no atmospheric transmission adjustments were required to the spectra. $CO_2$ and $H_2O$ absorption and emission regions could be seen in the measured spectra (e.g., at 1400, 1900 and 2500 nm in Fig. 2a) but were simply excluded from the model fitting process.

Since little thermal emission occurs in the HR1024i spectral range from targets below ~500 K, the approach is not very well suited to retrieving temperatures below this limit. As per Wooster et al. (2005), FRP estimates were derived as the sum of the radiative emissions from the retrieved flaming and smoldering components:

$$FRP = A \sum_{i}^{n=2} \sigma \, p_i \, T_i^4 \tag{8}$$

where $\sigma$ is the Stefan-Boltzmann constant $A$ is the FOV area (0.064 m$^2$). This shall be referred to as the FRP retrieved using the spectral-fitting method, $FRP_{SF}$.

Figure 2 shows an example modelled and measured spectra from a single experimental burn, with excellent agreement seen outside of the previously mentioned gaseous absorption and emission regions.

FRP retrievals made using the thermal infrared brightness temperatures from the PI400 appeared to show good agreement with those derived from the HR1024i VIS to SWIR spectra (see example in Fig. 3a). However, further validation investigations are required, such as in daytime retrievals. The data from the PI400 were subsequently used for all analysis

since they provide a higher measurement frequency, and one that matches those of the AKBD measures (see Sect. 3.3.3). The PI400 data also provides spatially mapped FRP data rather than just a single 'fire-integrated' value.

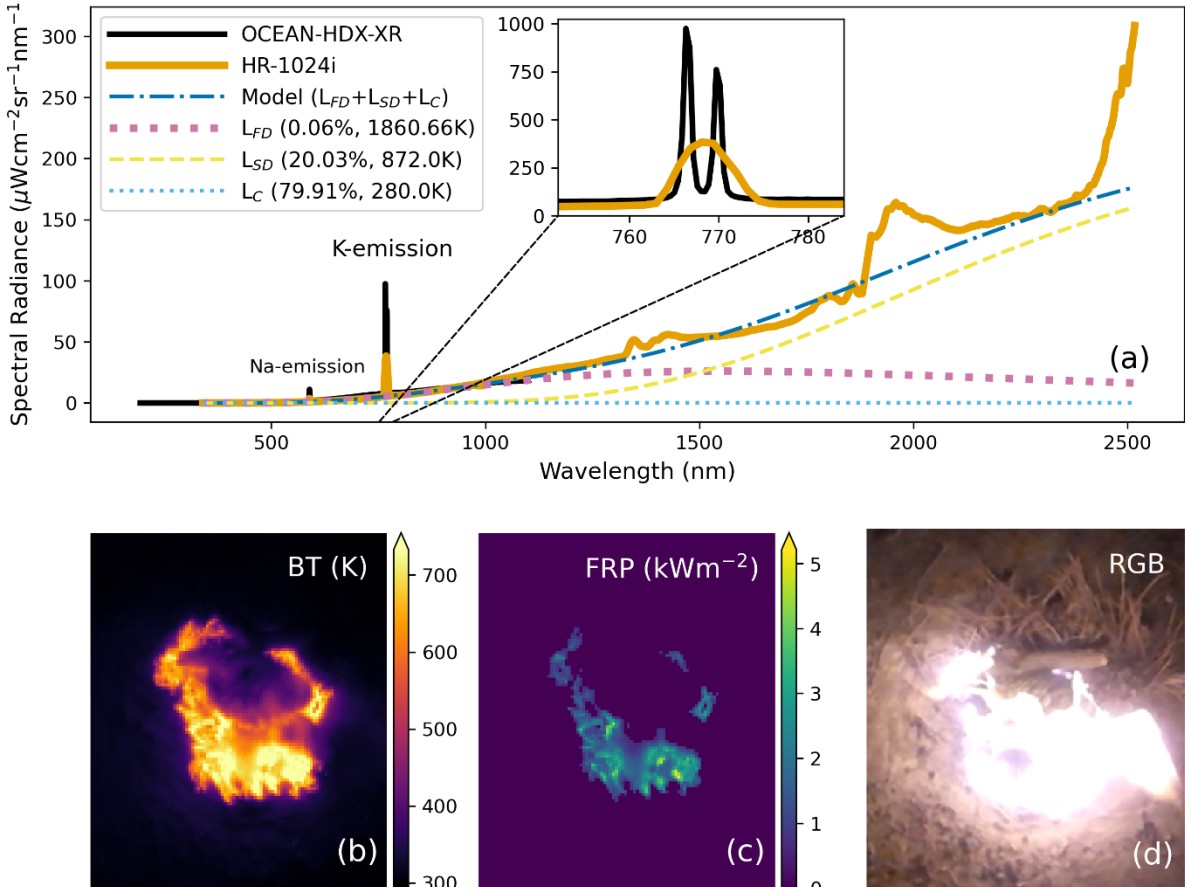

**Figure 2. Snapshot of an exemplar 23 x 23 cm experimental pine forest litter fire viewed from nadir using the setup shown in Fig. 1. Fire is shown at the time of maximum flaming activity: (a) Fire radiative emission spectra as measured by the OCEAN-HDX-XR (Ocean Insight, UV-to-NIR) and HR1024i (SVC, UV-to-SWIR) spectrometers, with magnitudes adjusted for the FOV difference between the two instruments. Modelled spectra derived from the spectral-fitting method described in Sect. 3.3.2 and its three temperature components (retrieved fractional area (%) and temperature (K) are listed in the legend); (b) Temperature (K) measured by the PI400 (Optris); (c) Fire Radiative Power (FRP, kWm$^{-2}$) derived from the PI400 thermal imagery, as calculated via the Stefan-Boltzmann law and omitting pixels below 650 K (see Section 3.3.1); (d) Frame from RGB video camera.**

**Figure 3. Data from the exemplar pine forest litter fire shown in Fig. 2: (a) FRP time-series derived from brightness temperature imagery collected by the Optris PI400 thermal imager and VIS-to-SWIR spectral collected by the**

**HR1024i. Data for a single time-step are shown in Fig. 2a; (b) Comparison of AKBD K-line metric derived from**

**spectra measured by the OCEAN-HDX-XR and HR1024i spectrometers around the location of the potassium emission line (see Sect. 3.3.3); (c) Time-series of excess (Δ) $CO_2$, CO and $CH_4$ (ppm) concentrations in the smoke from this fire, each normalized by their maximum value (given in the legend). The time flaming activity ceased as determined by the RGB video record is marked by the vertical dashed line.**

### 3.3.3 K-line measurements

An OCEAN-HDX-XR spectrometer (Table 2) was used to provide 1 Hz K-line measurements of each fire. The instrument measurement diameter was 53 cm at the fuel bed, and the instrument calibrated to provide data in spectral radiance units using an Ocean Insight HL-3P-CAL calibration lamp. The resulting spectra were used to measured K-line strength using the AKBD metric introduced in Sect. 2.

The HR1024i can also provide K-line spectra, though its 3.5 nm spectral resolution in the K-line region is coarser than the 1.1 nm of the OCEAN-HDX-XR. Comparison of their AKBD data (e.g., Fig. 3b) showed excellent agreement ($R^2$ of 0.99; with a linear best fit of gradient of 0.381±0.002 and negligible intercept). However, the AKBD measured by the OCEAN-HDX-XR are more than twice as large as from the HR1024i. This is despite the former's larger measurement area; this is due to its higher spectral resolution, which is evident in its ability to better resolve the individual potassium emission lines (see Fig. 2a). Since the OCEAN-HDX-XR also provided a higher measurement frequency, its AKBD data were used for all subsequent analysis.

### 3.4 Smoke measurements

### 3.4.1 $CO_2$, CO and $CH_4$ mixing ratios

Trace gas measures of the smoke from every fire was continuously directed via the hood and through the extraction flue (Fig. 1), and used to calculate smoke emission rates and MCE. Continuous $CO_2$, CO and $CH_4$ mixing ratio measures were taken at an inlet at location C (Fig. 1) using an adapted Los Gatos Research (LGR) Ultraportable Greenhouse Gas Analyzer laser absorption spectrometer (described in Zhang et al., 2015). However, for the crop residue fires, the LGR was unavailable, so a GasLab CM-1000 was deployed instead. This instrument uses a Non-Dispersive Infrared (NDIR) detector to assess $CO_2$ mixing ratios, and an electrochemical sensor to assess CO. The differing response times of these two detectors





were accounted for using the method of Zhang et al. (2020) such that continuous MCE measurements could be derived using

Eq. 1. Excess concentrations of $CO_2$, CO and $CH_4$, along with the matching FRP and AKBD data, are shown in Fig. 3c and

3d for an example pine forest litter fire.

### 3.4.2 Trace gas emission rates

To account for any variability in the extraction system, the gas flow rates through the flue were calculated by combining the

trace gas concentrations (Sect. 3.4.1), 1 Hz pressure difference measurements (dP [Pa]) between the inside and outside of the

flue using a Testo 440, and the flue gas temperature using two thermocouples. Gas velocity (v [m s$^{-2}$]) was derived from the

pressure data using the Bernoulli equation:

$$v = \sqrt{\frac{2d\mathrm{P}}{\rho_X}} \qquad (9)$$

where $\rho_X$ is the density of species X (kg m$^{-3}$). Temperatures of the flue gases varied by up to 40 K over the course of

individual fires. Therefore, $\rho_X$ was adjusted using:

$$\rho_X = \frac{MM_X P}{R T_{TC}} \qquad (10)$$

where $MM_X$ is the molar mass of gas species X, R is the ideal gas constant (8.3145 mol$^{-1}$K$^{-1}$), P is pressure, which was

assumed to be constant of 101 kPa as the pressure difference measurements from the Testo were negligible in comparison,

and $T_{TC}$ is the temperature recorded by a thermocouple in the flue (in K). The measured emission rate of X, $\left(\frac{dM_X}{dt}\right)_m$, in g s$^{-1}$

was then calculated by:

$$\left(\frac{dM_X}{dt}\right)_m = A_{CS}\rho_X v 10^{-6}\Delta X \qquad (11)$$

where $A_{CS}$ is the cross-sectional area of the flue at location D (0.01767 m$^2$) and $\Delta X$ is the excess concentration of X (in ppm)

in the flue and 10$^{-6}$ is a unit conversion factor for the gas concentration.

### 3.4.3 Experimental Procedure

In total, 12 pine forest litter, 12 oak kindling, and 8 crop residue fires were conducted over a period of 1 week, during which

ambient air temperature in the Chamber ranged from 9 to 12°C and relative humidity from 71 to 75%. After preparation of



each fuel bed, the extraction flue and all instruments were turned on and the pre-fire trace-gas ambient concentrations

calculated as the mean of the sixty 1 Hz concentration measurements taken immediately before each ignition. Ignition was

made using a blowtorch applied to one edge of the fuel bed, with a small amount of sawdust added to help ignite the oak

kindling. Measurements only ceased when concentrations of $CO_2$ and CO closely approached those pre-fire. Post-fire, the

cooling char, ash, and any unburned fuel was removed and the next fuel bed prepared. Fuel mass ranged from 125 g to 250 g

for the pine litter fires, 220 g to 410 g for oak, and 100 g to 200 g for crop residue. Fire duration across the 32 burns ranged

from 16 to 42 minutes.

## 4. Laboratory experiment results

### 4.1 K-line detection

As with previous work (Amici et al., 2011; Magidimisha and Griffith, 2017; Vodacek et al., 2002), the detection of a K-line

signature coincided with clear flaming combustion seen in the RGB video record. An AKBD $\geq$ 1.5 $\mu Wcm^{-2}sr^{-1}nm^{-1}$ was

always recorded by the OCEAN-HDX-XR when flames were visible. A higher MCE was also recorded when the AKBD

signal was high, indicating that a higher proportion of fuel carbon was being completely oxidized to $CO_2$ than when the

AKBD was low. We built on these observations to attempt to link the K-line emission signal to quantitative improvements in

our ability to estimate the trace gas emission.

### 4.2 Fire effective temperature and spectral fitting

Figure 4 shows the fire effective temperature and area parameters derived using fits to the spectra of a single pine forest litter

fire. Whilst the total FRP from these fits agrees with that derived using the PI400 as already stated (Fig. 3a), the individual

FRP calculated for the separate flaming and smoldering components do not appear very realistic. The highest combustion

rate is seen in the period from ignition to when all flaming activity ceased, and it is here that the highest rates of $CO_2$

production occur (Fig. 3c). However, the 'flaming FRP' is reported as far smaller than the 'smoldering FRP' even during this

period (Fig. 4a), even though fuel consumption was clearly dominated by flaming combustion at this time (also shown by the

$CO_2$ production in Fig. 3c). Therefore, these separate flaming and smoldering FRP components do not appear well matched

to how each combustion phase is actually contributing to overall fuel consumption.





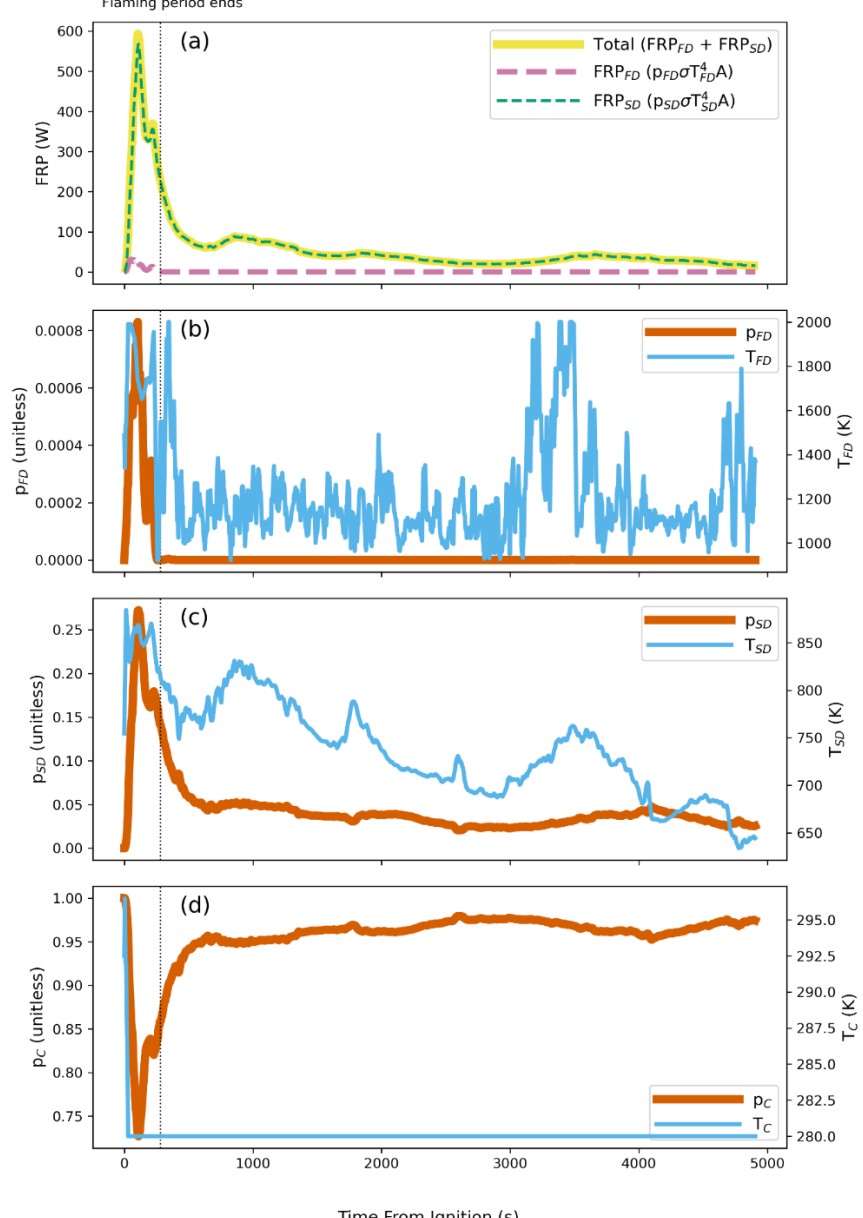

**Figure 4. Time-series of results of the spectral-fitting method for the pine forest litter fire also shown in Fig. 3: (a)**
**FRP derived from the two separate fire components (Eq. 6), the first is meant to represent the flaming zone (923 K <**
**$T_{FD}$ < 2000 K), the second the smoldering zone (623 K < $T_{SD}$ < 1023 K), and their sum is the total of these two; (b) The**
**fractional area, $p_{FD}$, and temperature, $T_{FD}$, of the first component representing flaming-dominated combustion; (c)**
**The fractional area, $p_{SD}$, and temperature, $T_{SD}$, of the second component representing smoldering-dominated**



**combustion; (d) The fractional area, $p_C$, and temperature, $T_C$, of the third component representing the cooling non-**

**fire background. The end of the flaming period, the time when all flames appeared to cease in the RGB camera data,**

**is indicated by the vertical dotted line.**

Going further, for all fires on which this method was tested, we found that retrievals of the second fire component derived

from the spectral fits and meant to represent the smoldering combustion contribution always dominated the total FRP. In

contrast, retrieved temperatures in the flaming zone ($T_{FD}$) were frequently very high, often above 1600 K during the flaming

period of Fig. 4b for example, but those of $p_{FD}$ (flaming proportional area) very small (e.g., never larger than 0.085%) –

which led to a low contribution to total FRP. As there is little emission from the cooler smoldering component in the spectral

region measured, this 'flaming' component is derived by fitting at shorter wavelengths and some of this signal is coming

from hot soot in the flames rather than from the burning fuel itself. Parameters such as flame depth and soot concentration,

which are significant drivers of flame emissivity (Johnston et al., 2014), therefore influence the retrievals. Similar to the

AKBD measurements, we observed that flames were present while $f_1 \geq 0.0016\%$ (Fig. 4a). However, any daytime

measurements will require the removal of the reflected solar radiation component present at VIS to SWIR wavelengths

(discussed further in Sect. 5.1.2). This task that can introduce large uncertainty, especially when dealing with course spatial

resolution imagery where at the shorter wavelengths the solar reflected signal maybe far larger than that from the fire. The

AKBD calculation is unaffected by this issue since it quantifies the strength of the K-line above the combination of reflected

solar radiation and Planckian emission signal.

### 4.3 Combustion phase emission relationships

Figures 5a, 5b and 5c show, respectively, how emission rates of $CO_2$, CO and $CH_4$ vary with FRP – in this case for the

example pine forest litter fire. Each data point is colored by MCE, indicating the mix of smoldering, and flaming combustion

ongoing at each measurement time. High MCE (maximally dominated by flaming combustion) is yellow, and lower MCE

(maximally dominated by smoldering combustion) is in dark blue. Points having a contemporaneous AKBD signal $\geq 1.5$

$\mu Wcm^{-2}sr^{-1}nm^{-1}$ are outlined, indicating the confirmed presence of flaming combustion via the K-line signal.

It is clear from the hysteresis-shaped patterns present in Fig. 5 that any assumption of a purely linear relationship between trace gas emission rate and FRP is flawed, and that instead such relationships are combustion phase dependent. This agrees

with previous lab-based studies comparing FRP to $CO_2$ and CO emission rates (e.g., Freeborn et al., 2008).

Generally, soon after ignition, the fires enter a 'flaming-dominated' stage that produces a steep linear increase in $CO_2$ production with increasing FRP (Fig. 5a). Smoldering can and does occur during this 'flaming-dominated' stage, but it is consuming very little of the fuel compared to flaming. This is different to the 'flaming period' – which we class as the period of the burn when flames are present as defined via AKBD thresholding. During this time, significant smoldering combustion

can also occur, particularly as the fire is transitioning away from the flaming-dominated stage. As the smoldering-dominated stage begins (no flaming activity), there also appears to be a linear relationship between FRP and $CO_2$ emission, but with a gradient far lower than that found during the flaming-dominated stage. During the flaming period, the data of Fig. 5a falls between the two linear clusters representing the flaming-dominated and smoldering-dominated stages, with MCE values indicating a mixed contribution from both flaming and smoldering activity.

As expected, for the 'preferentially smoldering' species CO (Fig. 5b) and $CH_4$ (Fig. 5c), the relationship between FRP and trace gas emission rates are opposite in nature to those of the 'preferentially-flaming' $CO_2$ (Fig. 5a). The emission rates of CO and $CH_4$ increase linearly with FRP, but with a slope significantly lower than that for $CO_2$ during the early flaming-dominated stage. During the smoldering-dominated stage however, these slopes increase but the data are less well fitted by a linear relationship than during the flaming-dominated stage. The latter is possibly contributed to by a lower plume buoyancy

during the smoldering-dominated stage, when some of the smoke may not have made it directly into the extraction flue.

Based on these data, we developed and tested three different empirical models that use the remotely sensed measures to estimate the fires trace gas emission rates. The first model uses only remotely sensed FRP measures - and supposes a 'fire-averaged' relationship between FRP and trace gas emission rate, which is assumed for example within the classical FREM approach developed for use with satellite FRP data (Mota and Wooster, 2018; Nguyen et al., 2023; Nguyen and Wooster,

2020). The second two models use the K-line measures in addition to FRP and are aimed at providing a more nuanced trace gas emission rate estimate by considering the different relations seen during flaming and smoldering combustion. We used data from 23 'training' fires across all fuel types to develop the parameters of each model. Nine were randomly selected

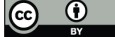

from each of the pine forest litter and oak kindling burns, and five from the crop residue burns. Three fires of each fuel type

were then used to evaluate, or 'test', the models.

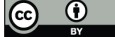

**Figure 5. Relationships between trace gas emission rate and FRP for (a) CO₂, (b) CO, (c) CH₄, and (d) total carbon**

**flux in moles (i.e., moles of CO₂, CO and CH₄). FRP here is derived from the PI400 brightness temperature imagery**

**as described in Section 3.3.1. These data are from a single pine forest litter training fire (Fire 11), and each point is**

**colored by its MCE at the time of measurement, and those when AKBD > 1.5 µWcm⁻²sr⁻¹nm⁻¹ (as derived from the**



**OCEAN-HDX-XR) indicating the presence of some flaming activity (flaming period) are distinguished from smoldering-dominated points by their outline. Emissions coefficients for flaming-dominated (MCE > 0.975), flaming-identified (AKBD > 1.5 µWcm$^{-2}$sr$^{-1}$nm$^{-1}$), fire-average and smoldering-dominated (AKBD < 1.5 µWcm$^{-2}$sr$^{-1}$nm$^{-1}$), which were calculated from the 'training' fires (including Fire 11) and are presented in Table 2.**

**4.4 Emission models account for combustion phase**

**4.4.1 Model 1: fire-average**

As with the FREM approach this model assumes a simple linear relationship between FRP and trace gas emission rate of species X, $\left(\frac{dM_X}{dt}\right)_{modelled}$.

$$\left(\frac{dM_X}{dt}\right)_{modelled} = C_A^X FRP \qquad (12)$$

where the average emission coefficient $C_A^X$, was calculated by dividing the sum of all emission rate measures by the sum of all FRP measures. This model acts as a baseline to which the performance of the further two models that incorporate K-line

information could be compared. $C_A^X$ values for the three fuels are presented in Table 2.

**Table 2: Emission coefficients, $C_{CPh}^X$, for the trace gas emission rates of $CO_2$ and CO (and $CH_4$ where measured), for each combustion phase (CPh) scenario (fire-average, flaming-dominated, flaming-identified, smoldering-dominated) for pine forest litter, oak kindling and crop residue fires. Units are g s$^{-1}$ MW$^{-1}$.**

| Trace Gas Emission Coefficients (g s$^{-1}$ MW$^{-1}$) | | | | | |
|---|---|---|---|---|---|
| | | Fire-average (A) | Flaming-dominated (FD) | Flaming-identified (FI) | Smoldering-dominated (SD) |
| Pine forest litter | $CO_2$ | 880 ± 2 | 1560 ± 12 | 1100 ± 4 | 523 ±2 |
| | CO | 33.4 ± 0.1 | 15.0 ± 0.1 | 25.8 ± 0.1 | 45.5 ± 0.1 |
| | $CH_4$ | 2.03 ± 0.01 | 0.586 ± 0.008 | 1.53 ± 0.005 | 2.82 ± 0.01 |
| Oak kindling | $CO_2$ | 888 ± 2 | 1950 ± 44 | 1030 ± 3 | 464 ± 2 |





| | | | | | |
|---|---|---|---|---|---|
| | CO | 22.8 ± 0.1 | 3.58 ± 0.08 | 18.0 ± 0.1 | 37.0 ± 0.1 |
| | CH$_4$ | 0.792 ± 0.002 | 0.0741 ± 0.002 | 0.749 ± 0.002 | 0.922 ± 0.003 |
| Crop residue | CO$_2$ | 804 ± 6 | 1670 ± 40 | 982 ± 10 | 434 ± 5 |
| | CO | 42.4 ± 0.4 | 16.4 ± 1.1 | 43.3 ± 0.6 | 40.5 ± 0.8 |

### 4.4.2 Model 2: FRP and AKBD magnitude (FAM)

In this second model, the emission rate of trace gas species X was modelled as the sum of that coming from separately considered flaming and smoldering activity:

$$\left(\frac{dM_X}{dt}\right)_{modelled} = C_{FD}^X FRP_{FD} + C_{SD}^X FRP_{SD} \tag{13}$$

where FRP$_{FD}$ and FRP$_{SD}$ are the contributions of the flaming-dominated (FD) and smoldering-dominated (SD) components of the fire to total FRP (i.e., FRP$_{FD}$ + FRP$_{SD}$ = FRP). $C_{FD}^X$ and $C_{SD}^X$ are the flaming-dominated and smoldering-dominated emission coefficients between FRP and the emission rate of species X. Values for $C_{FD}^X$ were derived by ratioing the total amount of species X emitted by the total FRP released for data during the time when MCE exceeded a threshold that produced the highest $R^2$ for a linear fit (e.g., MCE > 0.975 for pine). For the smoldering emission coefficients ($C_{SD}^X$) only data when AKBD < 1.5 µWcm$^{-2}$sr$^{-1}$nm$^{-1}$ were used since this confirmed the absence of flaming combustion. Examples are shown in Fig. 5, with the derived emissions coefficients for each fuel shown in Table 2.

This model requires distinguishing the relative contribution of each combustion phase to the total FRP recorded at any particular time. By assuming that FRP$_{FD}$ is directly proportional to AKBD (i.e. FRP$_{FD}$ = m$_k$ AKBD, where m$_k$ is a constant) Eq. 13 becomes:

$$\left(\frac{dM_X}{dt}\right)_{modelled} = C_{FD}^X\left(m_k\ AKBD\right) + C_{SD}^X\left(FRP - m_k\ AKBD\right) \tag{14}$$

where AKBD > 1.5 µWcm$^{-2}$sr$^{-1}$nm$^{-1}$ to exclude noise. m$_k$ values were determined by solving Eq. 14 with the measured AKBD, FRP and emission rates of the training fires using a least-squares approach. Mean and standard errors for the three fuels were: 4.99±0.04 W(µWcm$^{-2}$sr$^{-1}$nm$^{-1}$)$^{-1}$ (pine), 5.90±0.32 W(µWcm$^{-2}$sr$^{-1}$nm$^{-1}$)$^{-1}$ (oak) and 3.25±0.79 W(µWcm$^{-2}$sr$^{-1}$nm$^{-1}$)$^{-1}$ (crop residue). Although these standard errors do not overlap, the values of m$_k$ are rather close to one another, and this





was unexpected given that different fuels are likely to contain different concentrations of potassium. However, use of a single fuel-independent $m_k$ is very useful as it means any future application of the method would not need to know the fuel type that is burning. Therefore, we selected the mean $m_k$ of 4.71±0.28 W($\mu$Wcm$^{-2}$sr$^{-1}$nm$^{-1}$)$^{-1}$ for use with all fuels to in the

performance assessment stage, which was based on the nine test fires. Since this model relies on quantifying the strength of the potassium emission signal, we refer to it herein as the FRP and AKBD Magnitude (FAM) method or Model 2.

### 4.4.3 Model 3: FRP and AKBD identification (FAI)

As with the FAM model, this third model also considers the K-emission and FRP of the fire, but only uses the K-line presence or absence, rather than its magnitude. The rational for this is that NIR wavelength radiation maybe significantly

scattered by smoke in thick wildfire plumes, perhaps altering the measured AKBD magnitude. The model therefore does not attempt to separate $FRP_{FD}$ and $FRP_{SD}$, but instead multiplies the total FRP by different emission coefficients depending on whether or not a K-line is detected:

$$\left(\frac{dM_X}{dt}\right)_{modelled} = C_{FI}^X FRP, \text{ when AKBD} > 1.5 \ \mu\text{Wcm}^{-2}\text{sr}^{-1}\text{nm}^{-1} \tag{15}$$

$$= C_{SD}^X FRP, \text{ when AKBD} < 1.5 \ \mu\text{Wcm}^{-2}\text{sr}^{-1}\text{nm}^{-1} \tag{16}$$

where $C_{SD}^X$ is the same smoldering-dominated FRP-emission rate gradient from the FAM method in Eq. 13 for when no flaming is identified. Then, for when some flaming combustion is detected, FRP is instead multiplied by a 'flaming-

identified' ratio, $C_{FI}^X$. This was calculated by dividing the total emission of X by the total FRP when AKBD > 1.5 $\mu$Wcm$^{-2}$sr$^{-1}$nm$^{-1}$, i.e. during the flaming periods of the training fires. Examples for the three trace gases are shown in Fig. 5. The average across all fires for every fuel type tested are presented in Table 2.

### 4.5 Emission model evaluation and intercomparison

The performance of the three different remote sensing-based models in estimating trace gas emission rates were evaluated,

with results for one of the pine forest litter fires shown in Fig. 6. Figure 6a shows the FRP timeseries, and Fig. 6b the AKBD timeseries. Using combinations of these data, the conventional, fire-average Model 1 underestimates $CO_2$ production compared to reality during the early stages of a fire when flaming activity dominates, and then overestimates it during the subsequent smoldering stage. This can be seen in Fig. 6c, for example, and is a consequence of the relationship found



between FRP and $CO_2$ emission rate described in Sect. 4.3 and Fig. 5a. Likewise, the results for CO and $CH_4$ production act

in the opposite direction to those of $CO_2$, these being generally overestimated during the flaming-dominated stage and underestimated during the smoldering stage (see Supplementary Materials).

In addition to the total measured FRP, Fig. 6a also presents the estimated contribution of flaming and smoldering combustion to the FRP signal – based on the FAM (Model 2) approach. The FRP is initially totally dominated by that from flaming combustion, but variable amounts of smoldering combustion then commence and eventually become dominant. This pattern

appears much more realistic than the flaming and smoldering contributions to FRP calculated using the spectral-fitting approach (Sect. 4.2), which significantly underestimates flaming phase contributions to overall FRP. Model 2 also produces an estimated trace gas emission rate timeseries that is in far better agreement with the trace gas measures than are those from the fire-average Model 1. The comparison in Fig. 6d indicates that RMSE for the FAM model is less than half that of the fire-average approach, and the average RMSE reduction is reported in Table 3. For $CO_2$ these reductions were very

significant; 58±4% (pine); 46±4% (oak) and 35±13% (crop residue). For CO and $CH_4$ they are less significant, and the FAM (Model 2) approach showed little ability to improve upon the ability to estimate CO emission rates of the oak fires compared to Model 1 (-1.2±14.4 %). While the FAM approach produced a smaller instantaneous error for estimating CO than the fire-average approach for two of the three test fires, for the third the estimate was much poorer (see Fig. A6), which affected the average performance. However, there was some improvement for the other two fuel types, and in general Model 2 improves

CO and $CH_4$ emission rate estimation compared to Model 1 - but to a far lesser extent than for $CO_2$. However, for MCE the improvements provided by Model 2 over Model 1 are very significant, even for the oak fires. The reduction in RMSE compared to the fire-average approach ranges from 40±7% (soy) to 54±6% (pine).

Like the FAM approach, the FAI method (Model 3) also uses AKBD measures - but now just in a binary 'on/off' fashion, as detailed in Sect. 4.4.3. The approach proves significantly more accurate in terms of trace gas emission rate estimation than

does the 'fire-average' approach (Model 1) but does not perform as well as the FAM approach. For example, the reduction in RMSE for instantaneous MCE estimations is lower, ranging from 17±6% (soy) to 37±12% (pine).

Overall, our results from these combustion chamber burns indicate that FRP-based estimates of trace gas release rates can be greatly improved via the addition of K-line measurements. We also tested modified versions of the Model 2 (FAM) and 3





(FAI) approaches that estimated fuel combustion rate prior to trace gas emissions, but the performance was almost identical

to that without this additional step (see Appendix B). This work indicates that the relationship between FRP and combustion

rate differs significantly between smoldering and flaming combustion.

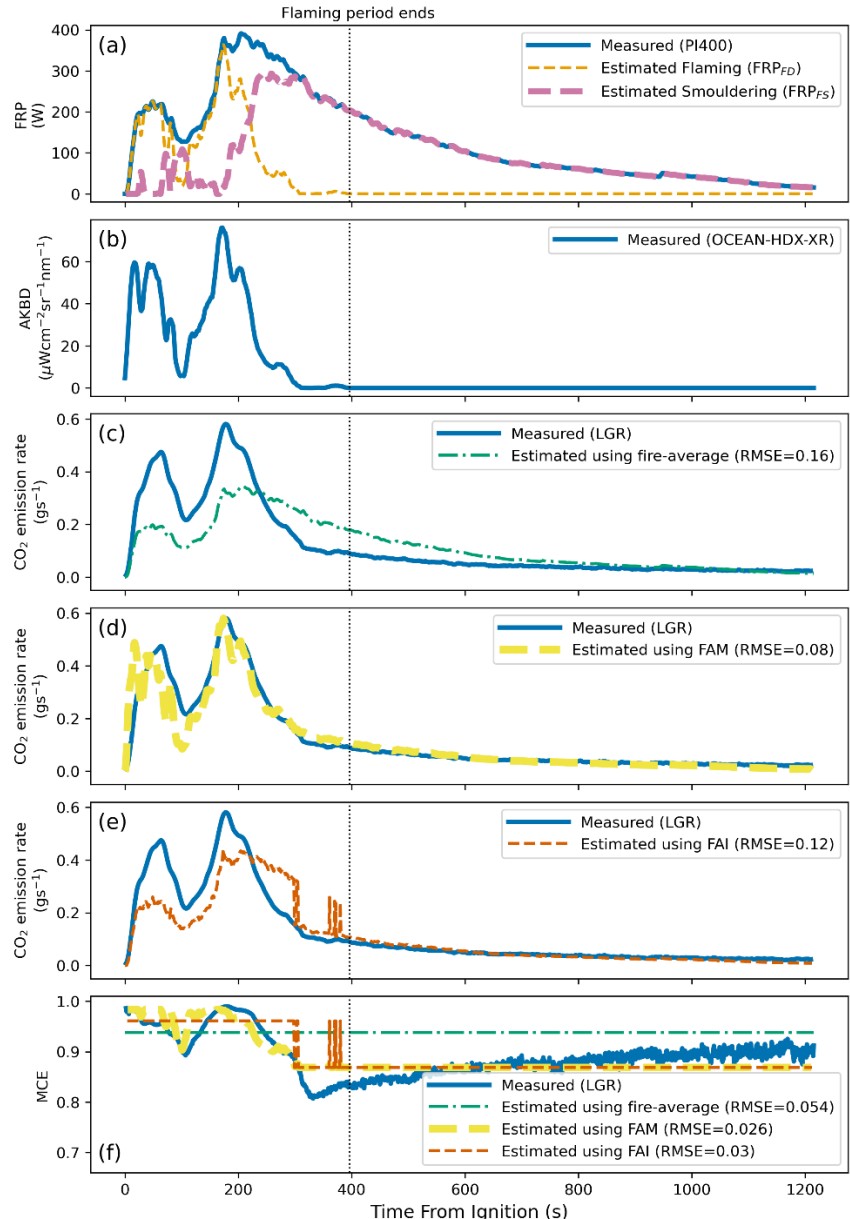

**Figure 6. FRP, AKBD and trace gas emission rate timeseries for an exemplar pine forest litter 'test' fire (Fire 3). (a)**

**FRP (W) calculated from the PI400 thermal imagery, along with the estimated flaming and smouldering components**





**derived from this and the complementary AKBD measures; (b) AKBD ($\mu Wcm^{-2}sr^{-1}nm^{-1}$) derived from spectra recorded by the OCEAN-HDX-XR spectrometer; trace gas emission rate of $CO_2$ ($gs^{-1}$) compared to that estimated with (c) the fire-average approach (Model 1); (d) the FAM approach (Model 2); (e) the FAI approach (Model 3); and (f) measured and modelled MCE. The end of the flaming period as determined by the RGB video camera record is denoted by the vertical dotted line.**


**Table 3: Mean RMSE between the measured and three modelled emission rates and MCE for each fuel type. Also includes the mean difference in RMSE for the FAM (Model 2) and FAI (Model 3) approaches compared to the fire-average (Model 1) method (%).**

| | | Approach Used to Estimate Emissions | | | | |
|---|---|---|---|---|---|---|
| Fuel | Trace Gas Species and MCE | Model 1: fire-average | Model 2: FAM | | Model 3: FAI | |
| | | Mean RMSE | Mean RMSE | Mean difference with fire-average (%) | Mean RMSE | Mean difference with fire-average (%) |
| Pine forest litter | $CO_2$ ($gs^{-1}$) | 0.18±0.02 | 0.10±0.02 | **-46±7** | 0.13±0.02 | **-27±3** |
| | CO ($mgs^{-1}$) | 3.9±0.3 | 3.5±0.5 | **-12±6** | 3.7±0.6 | **-6.1±9.2** |
| | $CH_4$ ($mgs^{-1}$) | 0.49±0.04 | 0.37±0.04 | **-26±2** | 0.44±0.02 | **-9.6±5.6** |
| | MCE | 0.048±0.003 | 0.022±0.002 | **-54±6** | 0.029±0.004 | **-37±12** |
| Oak kindling | $CO_2$ ($gs^{-1}$) | 0.45±0.14 | 0.24±0.08 | **-46±4** | 0.40±0.13 | **-14±3** |
| | CO ($mgs^{-1}$) | 6.1±0.5 | 6.1±1.1 | **-1.2±14.4** | 5.7±0.9 | **-8.8±7.5** |
| | $CH_4$ ($mgs^{-1}$) | 0.43±0.04 | 0.41±0.03 | **-4.9±1.3** | 0.43±0.06 | **-2.6±4.6** |





| | MCE | 0.056±0.003 | 0.032±0.001 | **-43±3** | 0.035±0.005 | **-37±7** |
|---|---|---|---|---|---|---|
| Crop residue | $CO_2$ $(gs^{-1})$ | 0.32±0.03 | 0.19±0.03 | **-35±13** | 0.28±0.02 | **-10±1** |
| | CO $(mgs^{-1})$ | 8.0±1.3 | 5.7±0.3 | **-23±12** | 7.0±0.6 | **-7.7±8.5** |
| | MCE | 0.052±0.007 | 0.030±0.001 | **-40±7** | 0.042±0.002 | **-17±6** |
| Average across all fuels | $CO_2$ | | | **-42±15** | | **-17±4** |
| | CO | | | **-12±20** | | **-7.5±14.6** |
| | $CH_4$ | | | **-15±5** | | **-6.1±7.2** |
| | MCE | | | **-46±10** | | **-30±15** |

**5. Applications to airborne data**

**5.1 Methodology**

**5.1.1 Airborne data**

The results in Sect. 4 show the benefits of combining remotely sensed FRP and AKBD measures when estimating trace gas emission rates of small-scale laboratory fires. However, ultimately, we aim at applications based on satellite EO measures and real landscape burning. Whilst satellite-derived K-line measurements from space have only been demonstrated once (Amici et al., 2012), the increasing launch of spaceborne imaging spectrometers provides the possibility for more routine observations with the necessary spatial/spectral resolution. Whilst we wait for those data, we here demonstrated the approach using airborne EO of real wildfires burning in the boreal forests of northern Ontario, Canada. The airborne remote data used comes from the Specim FENIX VIS-to-SWIR hyperspectral imager, covering the same spectral range with the same FWHM spectral resolution in the K-line region as the HR1024i, meaning we could apply essentially the same approaches and models developed in our laboratory study (Sect. 3.3.2 and 3.3.3).



### 5.1.2 FRP and K-line retrievals

Our FRP retrieval process required slight modification for the airborne EO application, in order to account for the reflected solar radiation present in the daytime imagery. For this, areas of water were masked from the scene and the active fire pixels detected using the HFDI index (Dennison and Roberts, 2009). All 'non-fire' pixels were then categorized into 'burned' and 'unburned' using the classification process of Waigl et al. (2019). Mean 'burned' and 'unburned' spectra were calculated using 200 pixels of each class, and these incorporated into the spectral-fitting model (Eq. 6) applied to each active fire pixel:

$$L_{\lambda,\text{ measured}} = p_u L_{\lambda,u} + p_b L_{\lambda,b} + \sum_i^{n=3} \varepsilon\,\tau\,p_i\,B_\lambda(T_i) \tag{17}$$

where $L_{\lambda,u}$ and $L_{\lambda,b}$ are the mean radiance calculated in the neighboring unburned and burned pixels at wavelength $\lambda$ and $p_u$ and $p_b$ are their area fractions of the pixel. All fractional areas sum to 1. The final component is the same as that used previously.

For inclusion in Eq. 17, the MODTRAN atmospheric radiative transfer model (Berk et al., 1999) was used to estimate atmospheric transmittance ($\tau$) from the surface to the aircraft altitude of 2500 m, assuming a 1976 US standard atmosphere, a 23 km visibility and an atmospheric $CO_2$ concentration of 420 ppm. As with the laboratory data, only wavelengths above 1200 nm were included in the spectral fitting approach and application of Eq. 16 to each active fire pixel, in order to reduce the influence of atmospheric scattering by smoke. Per-pixel FRP was again calculated from the derived $p_i$ and $T_i$ values using the Stefan-Boltzmann law (Eq. 7). Figure 7c shows measured and modelled spectra of a single example active fire pixel, and as with the HR1024i data (Fig. 2a) the two agree extremely well.

As this FRP retrieval method relies mostly on the second, smoldering temperature component ($L_{SD}$), it is likely to still be valid despite the unknown influence of reflected solar radiation at the smaller wavelengths on $L_{FD}$. Although further validation efforts are required to confirm this. On the other hand, relying on a non-zero value for the fractional area of $L_{FD}$ to indicate flaming combustion is no longer valid. The retrieved $p_{FD}$ for many pixels was often zero, whilst the magnitude of the AKBD parameter indicated flaming combustion was present. The opposite was also true: many pixels with a significant $p_{FD}$ had a very low or zero AKBD. Therefore, AKBD was more trusted than $p_{FD}$ for indicating the presence of flaming combustion since the measurement was not influenced by reflected sunlight.



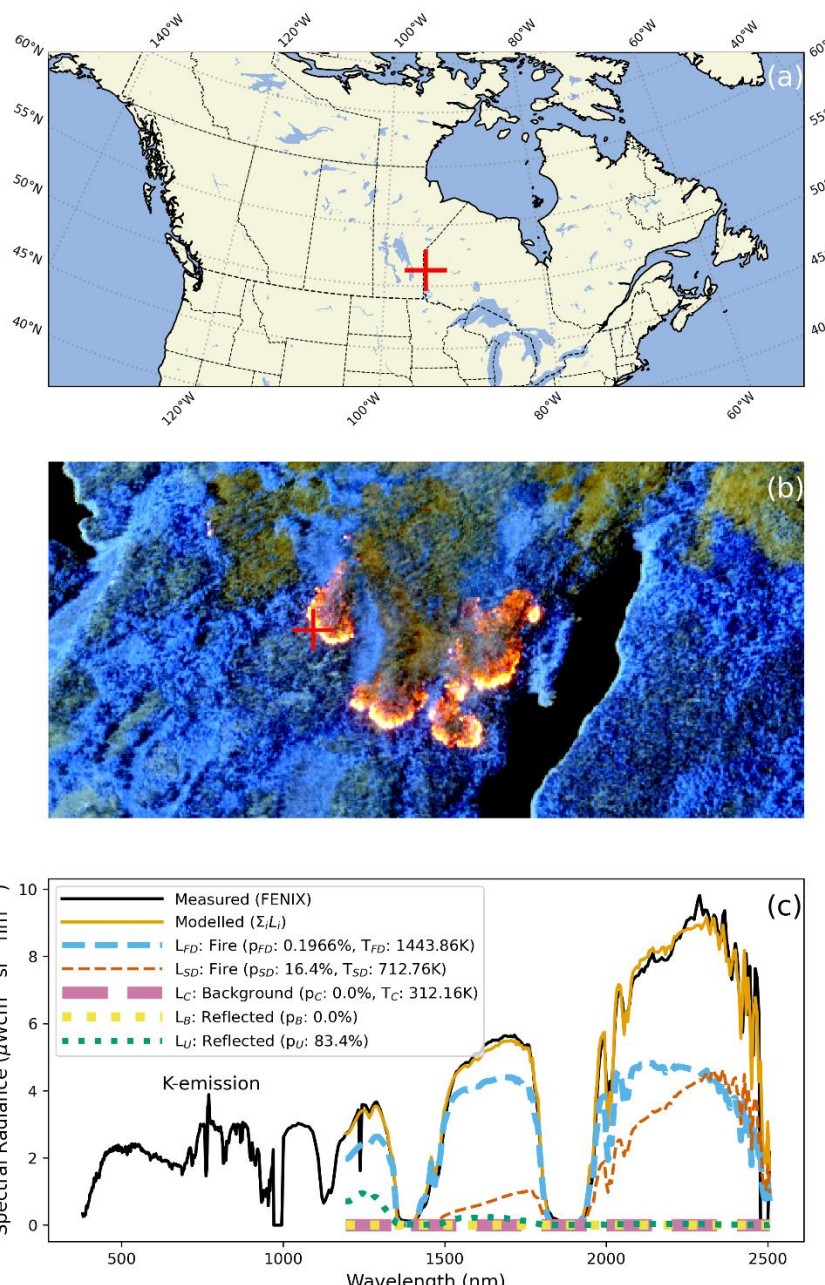


**Figure 7. Daytime airborne VIS-SWIR data taken on 11th August 2018 using a Specim FENIX hyperspectral imager flown over a boreal forest wildfire burning in Northwestern Ontario, Canada: (a) Map showing fire location (51.293° N, 94.805° W ); (b) Infrared color composite image comprised of data collected in bands centered at 2.2, 1.6 and 1.1 microns (RGB); (c) Exemplar spectra (VIS to SWIR) of an active fire pixel (location shown in the infrared**



**color composite (b)), with the best matched modelled spectra derived using the spectral-fitting approach (Eq. 16), along with the individual components that sum to that spectra: $L_{FD}$ (fire, flaming-dominated), $L_{SD}$ (fire, smoldering-dominated), $L_b$ (background/cooling fuel), and the reflected sunlight components $L_U$ (unburned) and $L_B$ (burned).**

The emission coefficients for the pine forest litter fuel, shown in Table 2, were used to convert the derived per-pixel FRP values to trace gas emission rates. For the FAM approach, the proportionality constant ($m_k$) linking AKBD and flaming FRP

(Eq. 13) was taken from the mean of $0.0201 \pm 0.0012$ W($\mu$Wcm$^{-2}$sr$^{-1}$nm$^{-1}$)$^{-1}$ determined for the three different fuels, with this value including an adjustment for differences between the pixel size and FWHM of the FENIX and HR1024i instruments. The same value was used in all results presented in Section 4 (and the Appendix).

### 5.2 Airborne data results

Figure 8a shows a true color composite of the wildfire, complementing the infrared color composite shown in Fig. 7. Thick

smoke blows away from the fire front, masking much of the land and supporting the decision to base FRP retrieval on wavelengths longer than 1200 nm less affected by Mie scattering (Sect. 5.1.2). By contrast, the longer NIR and SWIR wavelengths are far less affected, and the land and fire can be viewed clearly through the smoke (Fig. 8a). Figure 7b maps the FRP of each active fire pixel based on the spectral fitting method (see the example at a single pixel shown in Fig. 8b). Non-fire pixels are here colored depending on their proportion burned – $p_B$ and $p_U$ (Eq. 17). Overall, the imagery shows five

fire front heads having high-FRP and likely flaming activity advancing against the wind, with lower FRP areas in the (presumed largely smoldering) zone behind.

Figure 8c maps AKBD, used to class fire pixels as flaming or smoldering. The AKBD map indicates that flaming activity is indeed present in the high FRP fire front pixels, and only a few lower FRP spots of flaming activity exist in the largely smoldering zone. Applying the FAM approach, Fig. 8e, 8f and 8g show estimated $CO_2$, CO and $CH_4$ emission rates for the

fire, and Fig. 8h the MCE. Higher MCE smoke is emanating from the flaming fronts, and lower MCE from the smoldering zone apart from the few spots still flaming.

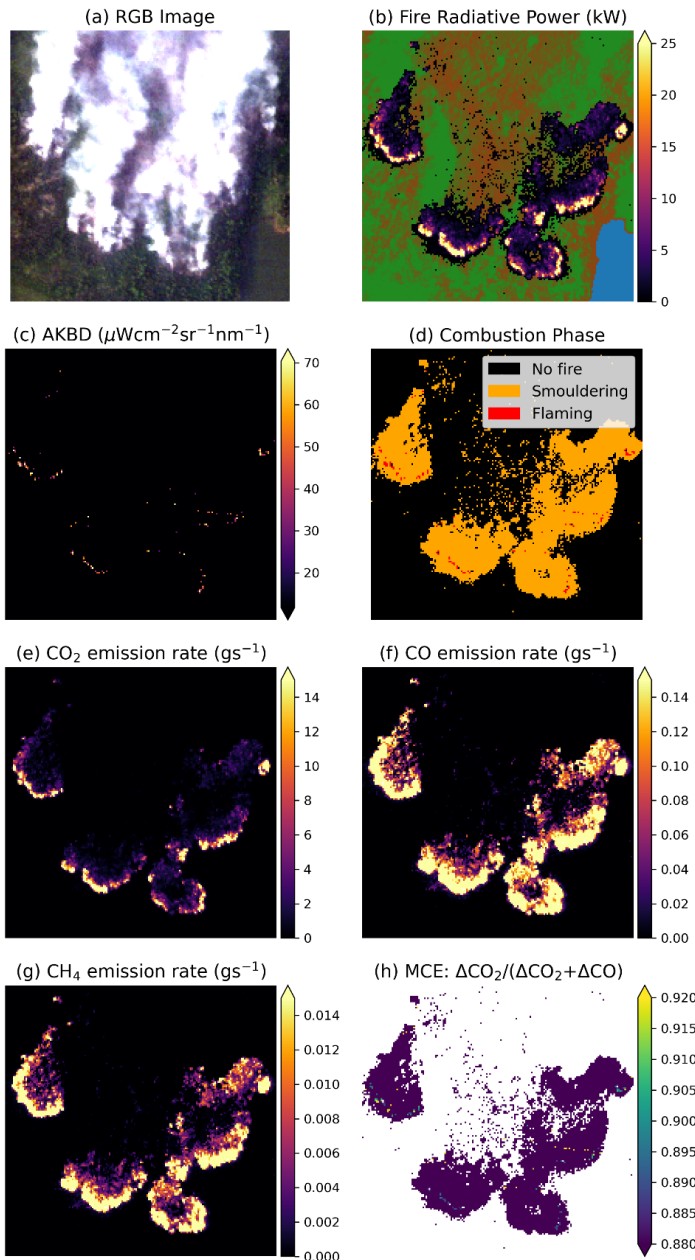

**Figure 8. 4 m spatial resolution airborne imagery and derived fire and smoke characteristics of the wildfire shown in Fig. 7. This scene is 700 m x 700 m wide. (a) RGB color composite to compare to the SWIR color composite shown in Fig. 7b; (b) FRP derived using spectral-fitting approach (Eq. 16), with non-fire pixels colored depending on whether they are classed as burned (brown) and unburned (green) via the method discussed in Sect. 5.1.2 , masked out water**

**pixels of a lake in the bottom right of the scene are shown in blue; (c) AKBD derived using Eq. 5 (Sect. 2); (d) Combustion phase of each active fire pixel derived from an AKBD threshold of 0.57 $\mu Wcm^{-2}sr^{-1}nm^{-1}$; emission rate derived using the FAM approach of (e) $CO_2$; (f) CO and (g) $CH_4$; along with (h) the MCE estimated using the same approach.**

The total emission rate of $CO_2$ for the entire fire is estimated using the FRP and AKBD data by the FAM (Model 2) as $20.2\pm0.1$ kgs$^{-1}$, with a relatively similar $22.9\pm0.2$ kgs$^{-1}$ estimated from Model 2 (FAI). Each is significantly lower than the $33.5\pm0.2$ kgs$^{-1}$ estimated using the Model 1 (FRP only) approach with the 'fire-averaged' emissions coefficient, essentially because this approach overestimates $CO_2$ emission rate when smoldering combustion is dominant (see Fig. 5a and 6c). The degree of overestimation could have been worse because only 1.4% of the detected active fire pixels in the airborne image of Fig. 8 contain flaming activity.

Since the majority of the fuel consumption is coming from smoldering combustion, the FAM and FAI approaches estimate higher emission rates for CO and $CH_4$ than does the fire-average approach. For CO the models estimate the following: $1270\pm10$ gs$^{-1}$ (fire-average), $1720\pm10$ gs$^{-1}$ (FAM) and $1630\pm10$ gs$^{-1}$ (FAI), and for $CH_4$: $77.2\pm0.6$ gs$^{-1}$ (fire-average), $107\pm1$ gs$^{-1}$ (FAM), and $100\pm1$ gs$^{-1}$ (FAI). Therefore, whilst flaming combustion contributes around 2.4% of total $CO_2$ production, it contributes only around 0.3% and 0.2% of CO and $CH_4$ production, respectively. Flaming combustion is therefore responsible for around 2% of the fuel consumption rate.

Since the wildfire smoke contained copious amounts of particulates (Fig. 8a), altered (and even missed) AKBD measurements may have resulted from Mie scattering. Therefore, this may make the FAI approach more appropriate than the FAM approach. Overall, while the trace gas emission rate estimates from the FAM and FAI approaches are relatively similar ($535\pm6$ and $592\pm9$ mols$^{-1}$ of carbon), they are very different to those of the fire-average approach ($811\pm10$ mols$^{-1}$ of carbon). Their application accounts for the fact that the fire is mostly smoldering – information that would otherwise be missed.

## 6. Summary and conclusions

Combustion phase and the proportion of flaming and smoldering activity occurring in landscape fires varies with fuel type and fuel characteristics. This, in turn, influences the rate of fuel consumption and smoke emission, along with the smoke

emissions chemical composition via its effect on the emissions factors (EFs) of the individual emitted chemical species. There is increasing interest in tailoring the EFs applied within fire emissions estimates, for example to cope with the presumed seasonal variations in emissions factors (Vernooij et al., 2023, 2022). Earth observation has long been suggested as a way to do this, by remotely mapping combustion phase to improve global smoke emissions estimation. We have

provided the first attempt at evaluating whether such methods actually lead to the desired improvements, using laboratory burns of three fuel types to test two approaches of determining combustion phase that (i) use remotely sensed retrievals of sub-pixel fire temperature, and (ii) utilization of potassium emission line (K-line) signatures that only occur during flaming combustion.

Whilst the first approach produced fire temperature estimates that were able to provide accurate FRP values, the individual

fit parameters were not easily related to the smoke emissions characteristics. On the other hand, measurements of the K-line emission strength were related to the emission rates of $CO_2$, CO and $CH_4$, leading us to develop two empirical models that used (Model 2) the magnitude of the K-line emission strength, and (Model 3) only its identification and combined these with the FRP data to better estimate fire smoke emissions rate and smoke chemical composition.

When compared to the standard FRP-only approach (Model 1) used to represent classical smoke emission estimate methods,

the combination of FRP and K-line data significantly increased the accuracy of the resulting emission rate estimates of the trace gases examined. It also significantly improved estimation of the modified combustion efficiency (MCE) of the fires smoke plumes, which directly relates to the contribution of flaming and smoldering combustion to the smoke. We did not find the ability to remotely determine sub-pixel fire effective temperature to provide similar improvements.

The FRP and K-line magnitude approach (Model 2) reduced the RMSE of the MCE estimates for the emitted smoke by

between 54±6% and 40±7%, depending on fuel type, when compared to the FRP only 'standard' approach (Model 1). The equivalent RMSE reductions seen for Model 3 were lower, but still significant (-37±12%, -37±7%, and -17±6%, respectively). These results provide evidence that complementary FRP and K-line data could be used widely to improve fire emissions estimation, not only for the gases tested here but also for many others since MCE is well correlated to the EF of many species (Bertschi et al., 2003; McMeeking et al., 2009; Urbanski, 2013; Yokelson et al., 1996). Our findings point to

the potential for using this method with future spaceborne high spectral resolution data that can map the K-line from space at

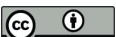

the same time as thermal remote sensing is used to retrieve the fires FRP. According to our findings, such complementary data shows for the first time a proven ability to determine sub-pixel fire combustion phase which can provide an ability to significantly improve fire emission estimation. We then demonstrate the value of the most effective approach on airborne remote sensing data of real wildfires, pointing the way to its ultimate application to potentially improving global fire

emissions estimation through its application to spaceborne observations.

**Data availability**

All the data in this study are available from the authors upon request.

**Author contributions**

FOB: Conceptualization, Methodology, Formal Analysis, Investigation, Writing – Original Draft, Visualization. MW:

Conceptualization, Funding acquisition, Writing – Review & Editing, Supervision. MG: Methodology, Writing – Review & Editing. YL: Resources, Writing – Review & Editing.

**Competing interests**

The contact author has declared that none of the authors have any competing interests.

**Acknowledgements**

We thank Dr Zixia Liu from King's College London for assisting with the collection techniques for the trace gas measurements in the laboratory. The Leverhulme Centre for Wildfires, Environment and Society and National Centre for Earth Observation (NCEO) are both thanked for their support of this research. We also thank the European Space Agency (ESA) for supporting the featured airborne campaign. NERC British Antarctic Survey (BAS) are thanked for their support in providing and operating the airborne platform used during the campaign, whilst the NERC Airborne Research and Survey

Facility (ARSF), and now the NCEO Airborne Earth Observatory (NAEO) Team of NCEO-King's College London, are thanked for coordinating and delivering the campaign. The UK's NERC Earth Observation Data Acquisition and Analysis



Service (NEODAAS) processed the hyperspectral airborne data used herein to a calibrated and co-located Level 1b product. The views in this publication can in no way be taken to reflect the official opinion of the European Space Agency or any other funding body.

**Financial support**

FOB is supported by a PhD Studentship from the London NERC DTP (grant no. NE/R012148/1). The practical work undertaken in this research was supported by the Leverhulme Centre for Wildfires, Environment and Society through the Leverhulme Trust (grant number RC-2018-023), by NERC National Capability funding to the National Centre for Earth Observation (NCEO, NE/Ro16518/1), and by the European Space Agency project 'Evaluation and Validation of Sentinel-3

Active Fire Detector and FRP Products experiment' (FIDEX; Contract number 4000136760/21/NL/FF/ab).

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

**Appendices**

**Appendix A – spectral -fitting model**

Reducing the modelled spectra to a single thermal component for the fire resulted in far poorer fits than with two

components, particularly at shorter wavelengths, and increasing the number of thermal components delivered results

having negligible fractional area and thus no detectable improvement in the modelled spectra.  Thus, two fire thermal

components were seen as the optimum choice – and allowed a high-quality fit to the measured spectra and a strong ability

to estimate total FRP, but the retrieved fire temperatures could not be used to reliably estimate extent of flaming and

smoldering activity.

**Appendix B – alternative approaches going via combustion rate**

As detailed in Section 1, most conventional remote sensing approaches to estimating fire emissions rely on first estimating

the fuel mass burned. Therefore, FAM and FAD approaches involving this step were also tested as an alternative to the

more direct approaches evaluated in Section 4.4. Combustion rate ($gs^{-1}$) was determined by first estimating total carbon

flux (Figure 5d) in moles and multiplying this by 12 $gmol^{-1}$, the resulting figure agreeing well with the mass loss obtained

from the scales data. Both emission factors and relationships between FRP and combustion rate were calculated for the

four combustion types: fire-average, flaming-dominated, flaming-identified and smouldering. Models analogous to FAM

and FAD were then tested. These estimated combustion rate, which was then multiplied by emission factors to give

emission rates. These were then compared to the measured rates. The results were almost identical to those presented in

Section 4.5 as the methods are also identical, except for the step of converting to combustion rate. This method did,

however, indicate that the relationship between measured FRP and rate of combustion was different between the two

combustion phases.
