# Peer review of "Can remote sensing combustion phase improve estimates of landscape fire smoke emission rate and composition?"

_Atmospheric Measurement Techniques, 2024_

## Author Comment (AC1)

We would like to thank the editor for handling our manuscript and the reviewer for providing constructive and supportive comments based on their reading of the paper. Please find our responses (regular text) to the reviewer's comments (**bold text**).

**(RC1) Line 95: Suggest changing "...the smoke production process:" to "...the smoke production process, defined as:"**

(AC) We have made this change to the revised manuscript.

**(RC1) Lines 105-108: A clarification of what "effective temperature" (T_f) actually means is warranted here since emissivity does not appear in Eqs. (2) and (3). This implies that T_f is a radiant temperatures rather than actual fire temperature as measured with a thermometer, but the intended meaning is not clear from the text.**

(AC) The reviewer makes a pertinent point. As they suggest, the earliest papers that include a "retrieved fire temperature" parameter refer to it as "radiant temperature," as it assumed an emissivity of one. An example can be found in Dozier (1981): https://doi.org/10.1016/0034-4257(81)90021-3. However, later papers began using either the term "effective temperature" or simply "temperature." We chose to use the former, as the latter term implies an actual surface temperature. The first example of the term "fire effective temperature" appears in Dennison & Matheson (2011): https://doi.org/10.1016/j.rse.2010.11.015, with a more recent instance in the review by Wooster et al. (2021): https://doi.org/10.1016/j.rse.2021.112694. We have made the clarification clear in the revised manuscript stating that these "fire effective temperatures" are, in fact, the estimates of the radiant temperature of the fire, rather than the kinetic temperature that might be measured with a thermometer. They assume that the fire is a grey body (e.g. https://doi.org/10.1071/WF12197) but make no assumptions with regards to the actual emissivity value.

**(RC1) Eq. 2 and 3: Please make mathematical notation here and in text consistent, e.g. within Eq. 2 and 3 the variable p is sometimes italicized and sometimes not.**

(AC) We have made this change to the revised manuscript.

**(RC1) Eq. 2 and 3: As the effective temperature temperature is define,**

(AC) Comment appears to be incomplete. However, we saw that while $T_r$ refers to retrieved effective temperature, this was not clear from the text and have clarified this in the revised manuscript, which this comment may be referring to.

**(RC1) Eqs. 4 and 17: Here also please use italics consistently for variables and constants - see comment above in reference to Eqs. 2 and 3.**

(AC) We have made this change to the revised manuscript.

**(RC1) Line 147: Suggest using Oxford comma for clarity: "oak kindling, pine forest litter, and soybean crop residue."**

(AC) We have made this change to the revised manuscript.

**(RC1) Line 163: Symbols/characters between "each image that had" and "600 K" seem to be garbled.**

(AC) Apologies, this was meant to read "T > 600 K." We have made this change to the revised manuscript.

**(RC1) Line 171: Suggest writing "(L_FD+L_SD+L_C)" as "(L_FD, L_SD, and L_C)" for clarity.**

(AC) We have made this change to the revised manuscript.

**(RC1) Line 369: Not clear what the uncertainty (0.28) attached to the mean m_k represents or how it was calculated.**

(AC) This is the propagated uncertainty of the values used to calculate the mean (i.e., from the three values and their standard uncertainties on Line 365) - we have made this clearer in the revised manuscript.

**(RC1) Eqs. (15): I think the condition here should be >= to be consistent with the K-line detection described in section 4.1.**

(AC) We have made this change to the revised manuscript.

**(RC1) Line 385: Here and later in text change "timeseries" to "time series".**

(AC) We have made this change to the revised manuscript.

**(RC1) Fig. 7: Please state approximate spatial dimensions of panel (b).**

(AC) The spatial resolution of Fig. 7b is 4 m and the size of the scene is 1020 x 2160 m. We have added this to the caption in the revised manuscript.

**(RC1) Fig. 8: Based on the panels above I would expect the MCE to show a bit more spatial variability. Would a nonlinear color scale possibly reveal more features?**

(AC) Thank you for the suggestion, and we agree that one might well expect more spatial variability in MCE. We did try several color scales to highlight this, but with no real success in revealing more

features than those that can be seen already (some flaming activity at the fire front with some small flaming spots behind in the smoldering zone).

The fire in Fig. 8 was mostly smoldering when the aircraft imaged it, as the K-line was only detected in 1.4% of the detected active fire pixels. This dominance of smouldering activity area explains the lack of MCE variability. We have mentioned this explicitly in the caption and in future we look forward to analysing new data of more flaming dominated wildfires.

**(RC1) Supplement Figs. S1-S11: Change "Times series" to "Time series" in captions.**

(AC) We have made this change to the revised manuscript.

---

## Author Comment (AC2)

We would like to thank the editor for handling our manuscript and the reviewer for providing constructive and supportive comments based on their reading of the paper. Please find our responses (regular text) to the reviewer's comments (**bold text**).

**(RC2) #line 21 – define FRP acronym**

(AC) We have defined this in Line 19.

**(RC2) #line 71 – '(Zhang et al., 2015)'**

(AC) We have corrected the missing opening bracket at the start of the citation in Line 75 in the revised manuscript.

**(RC2) #line 163 – typo - 'T ³ 600 K'**

(AC) Apologies – this was meant to read "T > 600 K." We have made this change to the revised manuscript.

**(RC2) #line 213 – 'measure'**

(AC) We have corrected this typo in the revised manuscript.

**(RC2) #Fig 2 – Closing bracket missing on plot y-axis**

(AC) We have corrected this typo in the revised manuscript.

**(RC2) #line 322 – define FREM acronym**

(AC) We have defined this in the revised manuscript

**(RC2) #Figure 5/line 301– for clarity it would be beneficial to include the percentage of observations which were detected as containing flaming combustion (e.g. AKBD > 1.5). It appears to be the majority in the plots although in reality most observations are in the smouldering phase (e.g. Fig 3) which has less variation in values.**

(AC) The reason that the smoldering observations do not appear to be dominating in number over the flaming-identified observations in Fig. 5 is because the points are stacked on top of each other. However, as you say, most points are in fact identifying the fire in the smoldering dominated stage – which you can see in other figures, such as Fig. 3. We cannot make the points any smaller in Fig. 5, otherwise the colors of the points (and therefore MCE) would not be visible. We agree that including the percentage of observations when the K-line was detected would be beneficial and have now included this in the updated Fig. 5 plot. Thank you for the suggestion.

**(RC2) #line 430 what is the spatial resolution of the hyperspectral data used in this analysis? To what extent does spatial resolution influence the detection of pixels containing flaming activity?**

(AC) The spatial resolution of the data of Fig. 7b is 4 m, and the size of the area covered by the scene is 1020 x 2160 m. We have made this clear in the caption in our revised manuscript.

The question with regards to the effects that spatial resolution of the hyperspectral data has on K-line detection is a good one.

For a constant spectral resolution, the magnitude of the K-line (AKBD) for a given amount of flaming combustion is expected to decrease linearly with increasing measurement area, as the signal from the flaming area starts not to fill the pixel and instead becomes a smaller proportion of the overall pixel area (which may include a surrounding non-flaming area and indeed non-fire background). Using this assumption, we adjusted for the differences in the field of view between the two spectrometers used in the lab study (SVC HR1024i and the Ocean Insight OCEAN-HDX-XR) by applying a linear adjustment factor based on the measurement area differences. This resulted in the measured radiances of the fires agreeing well in the wavelength overlap of the two spectrometers after the adjustment (except for at the K-line peaks due to differences in spectral resolution), as shown in Fig. 2a. Figure R1 below shows an expanded plot of the two spectra in Fig. 2a in the region where the wavelengths measured overlap.

[Figure]

*Figure R1. Emission spectra as measured by the OCEAN-HDX-XR (Ocean Insight) and HR1024i (SVC) at wavelengths where the two spectrometers overlap. The magnitudes have been adjusted for differences in measurement area between the spectrometers. Expanded view of Fig. 2a in the original manuscript.*

We have also included another plot (see Fig. R2) showing the higher spectral resolution data (OCEAN-HD-XR; FWHM: 1.1 nm) convolved to the spectral resolution of the lower spectral resolution data (HR1024i; FWHM: 3.5 nm). These, again, have been applied with the adjustment factor to account for the differences in measurement area. This shows that they agree well but with some minor differences that can be attributed to a couple of factors.

Firstly, the spectrometers had different measurement frequencies. The HR1024i had irregular acquisitions of between 4 and 7 sec separation. It was not clear which proportion of the 4 – 7 secs the spectra were being recorded vs. preparing for a new acquisition or processing the previous measurement. This was unlike the OCEAN-HDX-XR, which was constantly measuring the spectra, averaging every second. Therefore, we are unable to exactly convolve the OCEAN-HDX-XR data to that of the HR1024i temporally. However, as stated in the manuscript, AKBD retrievals from the spectrometers showed excellent agreement ($R^2$ of 0.99; with a linear best fit gradient of 0.381±0.002 – due to spectral resolution difference – and negligible intercept) when compared using the large amount of data we recorded over many experimental burns.

Another factor causing the difference is that the viewing geometries of the two spectrometers were slightly different as they could not physically be put any closer together, mainly due to the size of the HR1024i. This may have resulted in the 3-dimensional flames, which influence this region of the spectra more than the hot fuel, being viewed slightly differently.

[Figure]

*Figure R2. Emission spectra as measured by the OCEAN-HDX-XR (Ocean Insight) and HR1024i (SVC) from Fig. R1, but with the higher spectral resolution OCEAN-HDX-XR data (1.1 nm) convolved to the spectral resolution of the HR1024i data (3.5 nm).*

In summary, after the adjustment factor has been applied the two measurements agree well but with some minor differences that can be expected. For this reason, we believe that our assumption above about how spatial resolution affects the AKBD retrieval is correct.

The flaming detection threshold of 1.5 $\mu Wcm^{-2}sr^{-1}nm^{-1}$ used in the analysis of the laboratory experiment refers to AKBD measured by the OCEAN-HDX-XR. This is equivalent to 0.57 $\mu Wcm^{-2}sr^{-1}nm^{-1}$ for both the HR1024i and the FENIX (from the airborne study), due to their different spectral resolution. This threshold was above the noise level of all the instruments.

For the airborne demonstration, the situation is rather similar to the lab fires. For instance, the airborne spectral fit indicated that the fire filled just over 20% of the measurement area of the laboratory VIS-SWIR spectrometer in the situation shown in Fig. 2a, and around 16% of the pixel in the airborne data example shown in Fig. 7c. This is due to the high spatial resolution that can

be achieved with airborne data, meaning that the fire fills a considerable amount of each pixel, to a similar extent as in the laboratory study.

However, future work to apply these findings to satellite data would be expected to account for lower spatial resolution, meaning fire will likely fill a smaller proportion of pixels. Therefore, the threshold may have to be lowered. The effect of spatial resolution, spectral resolution, the noise of the instrument, and how these factors influence the detection threshold will all have to be considered for each particular application.

**(RC2) References – a couple of references are missing details (e.g.  Magidimisha et al and  Urbanski, S.)**

(AC) We have corrected these in the revised manuscript.

**(RC2) Supplementary material  - In some plots the legend would benefit from being repositioned or reducing the text in the legend to avoid overlapping the data**

(AC) We have made changes (reducing text and/or moving legends out of the axes) in the revised manuscript to reduce the covering of data with the legends, which is particularly an issue in the MCE plots.

---

## Author Response (AR1)

Thank you very much for handling our manuscript. Please find our responses and changes made for each comment from the reviewers below in the following structure:

**Reviewer comment**

    A. Author response
    B. Changes to the revised manuscript (Version 2) from the original manuscript (Version 1)

These are in the order that they are read in the revised manuscript (Version 2).

**(RC2) #line 21 – define FRP acronym**

    A. (AC) We have defined this in Line 19.
    B. No changes made.

**(RC2) #line 71 – '(Zhang et al., 2015)'**

    A. (AC) We have corrected the missing opening bracket at the start of the citation in Line 75 in the revised manuscript.
    B. Changed "Zhang et al., 2015)" on Line 75 in Version 1 to "(Zhang et al., 2015)" on Line 75 in Version 2.

**(RC1) Line 95: Suggest changing "...the smoke production process:" to "...the smoke production process, defined as:"**

    A. (AC) We have made this change to the revised manuscript.
    B. Changed "the smoke production process:" on Line 95 in Version 1 to "the smoke production process, defined as:" on Line 95 in Version 2.

**(RC1) Eq. 2 and 3: Please make mathematical notation here and in text consistent, e.g. within Eq. 2 and 3 the variable p is sometimes italicized and sometimes not.**

    A. (AC) We have made this change to the revised manuscript.
    B. Italicised all variables and physical constants in the text between Lines 108 – 184 in Version 2.

**(RC1) Eq. 2 and 3: As the effective temperature temperature is define,**

    A. (AC) Comment appears to be incomplete. However, we saw that while $T\_r$ refers to retrieved effective temperature, this was not clear from the text and have clarified this in the revised manuscript, which this comment may be referring to.

    B. Changed "Temperature T" on Line 114 in Version 1 to "retrieved temperature $T\_r$" on Line 114 in Version 2.

**(RC1) Eqs. 4 and 17: Here also please use italics consistently for variables and constants - see comment above in reference to Eqs. 2 and 3.**

    A. (AC) We have made this change to the revised manuscript.

    B. Italicised all variables and physical constants in the text between Lines 108 – 184 in Version 2.

**(RC1) Lines 105-108: A clarification of what "effective temperature" ($T\_f$) actually means is warranted here since emissivity does not appear in Eqs. (2) and (3). This implies that $T\_f$ is a radiant temperatures rather than actual fire temperature as measured with a thermometer, but the intended meaning is not clear from the text.**

    A. (AC) The reviewer makes a pertinent point. As they suggest, the earliest papers that include a "retrieved fire temperature" parameter refer to it as "radiant temperature," as it assumed an emissivity of one. An example can be found in Dozier (1981): https://doi.org/10.1016/0034-4257(81)90021-3. However, later papers began using either the term "effective temperature" or simply "temperature." We chose to use the former, as the latter term implies an actual surface temperature. The first example of the term "fire effective temperature" appears in Dennison & Matheson (2011): https://doi.org/10.1016/j.rse.2010.11.015, with a more recent instance in the review by Wooster et al. (2021): https://doi.org/10.1016/j.rse.2021.112694. We have made the clarification clear in the revised manuscript stating that these "fire effective temperatures" are, in fact, the estimates of the radiant temperature of the fire, rather than the kinetic temperature that might be measured with a thermometer. They assume that the fire is a grey body (e.g. https://doi.org/10.1071/WF12197) but make no assumptions with regards to the actual emissivity value.

    B. Added "Note that "effective temperature" refers to an estimate of the radiant temperature of the fire, rather than the kinetic temperature one might measure with a thermometer. This assumes that the fire is a grey body (Johnston et al., 2014) but makes no assumptions with regards to the actual emissivity value." To Lines 116-119 In Version 2. Note that the reference was already in the references section as it is cited later in the text.

**(RC1) Line 147: Suggest using Oxford comma for clarity: "oak kindling, pine forest litter, and soybean crop residue."**

    A. (AC) We have made this change to the revised manuscript.

    B. Changed "oak kindling, pine forest litter and soybean crop residue" on Line 147 in Version 1 to "oak kindling, pine forest litter, and soybean crop residue" on Line 149 in Version 2.

**(RC1) Line 163: Symbols/characters between "each image that had" and "600 K" seem to be garbled.**

&

**(RC2) #line 163 – typo - 'T $^3$ 600 K'**

    A.  (AC) Apologies – this was meant to read "T > 600 K." We have made this change to the revised manuscript.

    B.  Changed "T $^3$600 K" on Line 163 in Version 1 to "T > 600 K" on Line 165 in Version 2.

**(RC1) Line 171: Suggest writing "(L_FD+L_SD+L_C)" as "(L_FD, L_SD, and L_C)" for clarity.**

    A.  (AC) We have made this change to the revised manuscript.

    B.  Changed "(L_FD+L_SD+L_C)" on Line 171 in Version 1 to  "(L_FD, L_SD, and L_C)" on Line 173 in Version 2.

**(RC2) #Fig 2 – Closing bracket missing on plot y-axis**

    A.  (AC) We have corrected this typo in the revised manuscript.

    B.  Replaced Figure 2 (line 193 in Version 1, line 195 in Version 2) to include closing bracket on y axis of (a) and correct the scale fo the y axis on the subfigure in (a).

**(RC2) #line 213 – 'measure'**

    A.  (AC) We have corrected this typo in the revised manuscript.

    B.  Changed "measured" on Line 213 in Version 1 to "measure" in Line 215 in Version 2.

**(RC2) #line 322 – define FREM acronym**

    A.  (AC) We have defined this in the revised manuscript

    B.  Changed "FREM" on Line 323 in Version 1 to "Fire Radiative Energy Emissions (FREM)" on Lines 325-326 in Version 2.

**(RC2) #Figure 5/line 301– for clarity it would be beneficial to include the percentage of observations which were detected as containing flaming combustion (e.g. AKBD > 1.5). It appears to be the majority in the plots although in reality most observations are in the smouldering phase (e.g. Fig 3) which has less variation in values.**

    A.  (AC) The reason that the smoldering observations do not appear to be dominating in number over the flaming-identified observations in Fig. 5 is because the points are stacked on top of each other. However, as you say, most points are in fact identifying the fire in the smoldering dominated stage – which you can see in other figures, such as Fig. 3. We cannot make the points any smaller in Fig. 5, otherwise the colors of the points (and therefore MCE) would not be visible. We agree that including the percentage of

observations when the K-line was detected would be beneficial and have now included this in the updated Fig. 5 plot. Thank you for the suggestion.

B. Added % of points where flaming was and was not identified to the legend of Fig. 5 on Line 332 of Version 2.

**(RC1) Line 369: Not clear what the uncertainty (0.28) attached to the mean m_k represents or how it was calculated.**

A. (AC) This is the propagated uncertainty of the values used to calculate the mean (i.e., from the three values and their standard uncertainties on Line 365) - we have made this clearer in the revised manuscript.

B. Added ", along with its propagated uncertainty, " to Line 371 of Version 2.

**(RC1) Eqs. (15): I think the condition here should be >= to be consistent with the K-line detection described in section 4.1.**

A. (AC) We have made this change to the revised manuscript.

B. Changed ">" to "≥" in Equation 15 on Line 381 of Version 2.

**(RC1) Line 385: Here and later in text change "timeseries" to "time series".**

A. (AC) We have made this change to the revised manuscript.

B. Changed "timeseries" on both Lines 384 and 385 in Version 1 to "time series" on Lines 388 and 389 in Version 2.

**(RC1) Eqs. 4 and 17: Here also please use italics consistently for variables and constants - see comment above in reference to Eqs. 2 and 3.**

A. (AC) We have made this change to the revised manuscript.

B. Italicised all variables and physical constants in the text between Lines 451 – 458 in Version 2.

**(RC1) Fig. 7: Please state approximate spatial dimensions of panel (b).**

&

**(RC2) #line 430 what is the spatial resolution of the hyperspectral data used in this analysis? To what extent does spatial resolution influence the detection of pixels containing flaming activity?**

A. (AC) The spatial resolution of Fig. 7b is 4 m and the size of the scene is 1020 x 2160 m. We have added this to the caption in the revised manuscript. Please see our detailed reply to RC2's question on spatial resolution in our Author's reply.

B. "with spatial resolution of 4m, the scene is 1020 m x 2160 m" added to the caption of Fig. 7 on Line 472 of Version 2.

**(RC1) Fig. 8: Based on the panels above I would expect the MCE to show a bit more spatial variability. Would a nonlinear color scale possibly reveal more features?**

A. (AC) Thank you for the suggestion, and we agree that one might well expect more spatial variability in MCE. We did try several color scales to highlight this, but with no real success in revealing more features than those that can be seen already (some flaming activity at the fire front with some small flaming spots behind in the smoldering zone). The fire in Fig. 8 was mostly smoldering when the aircraft imaged it, as the K-line was only detected in 1.4% of the detected active fire pixels. This dominance of smouldering activity area explains the lack of MCE variability. We have mentioned this explicitly in the caption and in future we look forward to analysing new data of more flaming dominated wildfires.

B. No changes made.

**(RC2) References – a couple of references are missing details (e.g. Magidimisha et al and Urbanski, S.)**

A. (AC) We have corrected these in the revised manuscript.

B. Journal name "International Journal on Advances in Software" added to Magidishima et al. (2023) on Line 651 in Version 2, but we could not find a DOI. " 317, 51–60. https://doi.org/10.1016/j.foreco.2013.05.045" added to Urbanski (2014) reference on Line 684 in Version 2.

**(RC2) Supplementary material  - In some plots the legend would benefit from being repositioned or reducing the text in the legend to avoid overlapping the data**

A. (AC) We have made changes (reducing text and/or moving legends out of the axes) in the revised manuscript to reduce the covering of data with the legends, which is particularly an issue in the MCE plots.

B. Moved legend in some of the plots in the supplementary materials and changed the legend text to Model 1, Model 2, and Model 3.

**(RC1) Supplement Figs. S1-S11: Change "Times series" to "Time series" in captions.**

A. (AC) We have made this change to the revised manuscript.

B. Corrected in the captions of all supplementary materials figures.

---

## Author Response (AR2)

Thank you very much for handling our manuscript. It has been a very positive experience. The peer review process was constructive and supportive.

We have changed the Data Availability statement on Lines 555-556 to:

*The data supporting this article are openly available from the King's College London research data repository, KORDS, at https://doi.org/10.18742/26826448*

We have also included the data citation in the Reference section on Lines 670-671.